# CLIPin: A Non-contrastive Plug-in to CLIP for Multimodal Semantic Alignment

## Abstract

Large-scale natural image-text datasets, especially those automatically collected from the web, often suffer from loose semantic alignment due to weak supervision, while medical datasets tend to have high cross-modal correlation but low content diversity. These properties pose a common challenge for contrastive language-image pretraining (CLIP): they hinder the model's ability to learn robust and generalizable representations. In this work, we propose **CLIPin**, a unified non-contrastive plug-in that can be seamlessly integrated into CLIP-style architectures to improve multimodal semantic alignment, providing stronger supervision and enhancing alignment robustness. Furthermore, two shared pre-projectors are designed for image and text modalities respectively to facilitate the integration of contrastive and non-contrastive learning in a parameter-compromise manner. Extensive experiments on diverse downstream tasks demonstrate the effectiveness and generality of CLIPin as a plug-and-play component compatible with various contrastive frameworks. Code is available at [Anonymous URL].

## 1 Introduction

CLIP has shown remarkable success in learning joint representations from large-scale image-text pairs, delivering strong performance across a wide range of downstream tasks in both natural and medical domains (Radford et al., 2021; Jia et al., 2021; Goel et al., 2022; Zhang et al., 2022b; Huang et al., 2021; Du et al., 2024). Despite its effectiveness, CLIP often suffers from inherent challenges in image-text datasets. Specifically, many large-scale natural image-text datasets used in CLIP-style pretraining (Thomee et al., 2016; Sharma et al., 2018; Schuhmann et al., 2021) are automatically crawled from the web with minimal or no human supervision, resulting in loose or inaccurately aligned pairs. This semantic noise undermines effective cross-modal representation learning by introducing ambiguity (Zhou et al., 2023; Li et al., 2021a; 2022b; Jia et al., 2021; Wu et al., 2022). For medical datasets, they usually exhibit accurate alignment, since the reports are written by clinicians based on image readings. However, the diversity of textual descriptions is limited due to the small variety of diseases and anatomical variations. In these cases, the CLIP often suffers from semantically similar samples being treated as negative sample pairs (negatives) (Yang et al., 2024; Wang et al., 2022). These two issues differ in form: semantic looseness reflects ambiguous positives (fuzzy one-to-many alignment), while semantic redundancy reflects homogeneous negatives (structured repetition within a batch). Despite this, they both violate the core assumption of the InfoNCE loss (Oord et al., 2018), namely that each positive pair is surrounded by mutually exclusive negatives. As a result, the model supervision becomes noisy or ambiguous, ultimately impairing the quality of learned representations.

Prior works have attempted to enhance representation quality under these limitations by introducing architectural modifications and multi-task objectives, such as incorporating image-text matching (ITM) losses and cross-modal attention mechanisms (Li et al., 2021a; 2022b). While these methods introduce complex and effective constraints, they remain grounded in the contrastive learning paradigm, thus inherit its limitations. Other approaches have incorporated non-contrastive components to improve inter-modal alignment and intra-modal diversity from a distributional perspective (Zhou et al., 2023). However, they typically lack explicit modeling of fine-grained, instance-level semantic correspondence.

To address these challenges, we propose **CLIPin**, a unified plug-in that enables non-contrastive feature representation to integrate with CLIP-style architectures, to enhance multimodal representation learning within image-text pretraining paradigms. Our key contributions are as follows: (*i*) We introduce a general and modular non-contrastive strategy that can be seamlessly integrated into existing contrastive frameworks without modifying their base architectures. By leveraging two semantically consistent yet independently augmented views per sample, our approach enables diverse and robust representation learning through distinct pathways without additional supervision. (*ii*) We design two shared pre-projectors for image and text modalities respectively, for facilitating the integration of contrastive and non-contrastive branches in a parameter-compromise manner. (*iii*) Extensive experiments across a wide range of downstream tasks, demonstrate that CLIPin consistently improves feature quality and cross-modal alignment, while serving as a plug-and-play module with strong generalizability across various contrastive architectures.

## 2 RELATED WORK

**Contrastive language-image pretraining.** Contrastive learning was first established in single-modal representation learning, particularly in vision tasks. Methods such as Caron et al. (2021); Oquab et al. (2024); Chen et al. (2020); Caron et al. (2020); Li et al. (2021b) have achieved impressive performance by contrasting different augmented views of the same image and learning inter-instance discrimination. Despite its simplicity and effectiveness, contrastive learning still faces practical challenges, particularly its heavy reliance on both the quantity and quality of negative sample pairs. On the one hand, effective estimation of the InfoNCE objective requires large batch sizes, which imposes significant memory and hardware demands. On the other hand, the representativeness and semantic diversity of negative sample pairs are crucial, unrepresentative or semantically similar negatives can reduce alignment precision and impair training. To address these limitations, methods like MoCo (He et al., 2020) introduce a memory bank and momentum encoder to decouple batch size from the number of negatives. Other approaches, such as PCL (Li et al., 2021b) and SwAV (Caron et al., 2020), employ clustering to avoid semantically redundant negatives, thereby improving training stability and representation quality.

Building on the success of vision-only models, contrastive learning has become a dominant paradigm in multimodal representation learning, with CLIP (Radford et al., 2021) as a representative framework. CLIP adopts a dual-encoder architecture trained with InfoNCE loss to align image and text representations in a shared embedding space. By pulling features of paired samples together and pushing mismatched ones apart, CLIP enables significant performance across diverse downstream tasks in both natural and medical domains. To improve robustness in the multimodal setting, recent works have augmented contrastive frameworks with auxiliary objectives (e.g., image-text matching, masked language modeling, or caption generation) and architectural refinements such as momentum encoders and query-based transformers (Li et al., 2021a; 2022b; 2023; Yu et al., 2022).

**Non-contrastive learning for feature representation.** Non-contrastive learning offers a compelling alternative by eliminating the need for negative sample pairs (Grill et al., 2020; Chen & He, 2021; Zbontar et al., 2021; Jing et al., 2022; Wen & Li, 2022). Methods such as SimSiam (Chen & He, 2021) and BYOL (Grill et al., 2020) achieve representation learning by encouraging consistency between positive pairs (e.g., different augmentations of the same sample) using an online-target architecture, where the target network is updated via exponential moving average (EMA). These approaches have shown strong performance in single-modal tasks, but their adoption in multimodal settings remains limited, because non-contrastive methods are highly sensitive to the interplay between model capacity and data scale, relying heavily on strong augmentations, and requiring careful design to avoid representation collapse (Li et al., 2022a; Wetzer et al., 2023; Vahidi et al., 2024; Huang et al., 2024; Wen & Li, 2022; Zhang et al., 2022a). In multimodal contexts, where image and text encoders are inherently heterogeneous, these issues are further amplified.

Until now, only xCLIP (Zhou et al., 2023) has attempted to extend non-contrastive learning to vision-language settings, which aligns the output distributions of the image and text encoders by optimizing both their sharpness and smoothness. However, its non-contrastive component focuses solely on batch-level distribution alignment and lacks explicit modeling of instance-level semantic correspondence. Furthermore, its training objective is decoupled from CLIP-style representation learning,

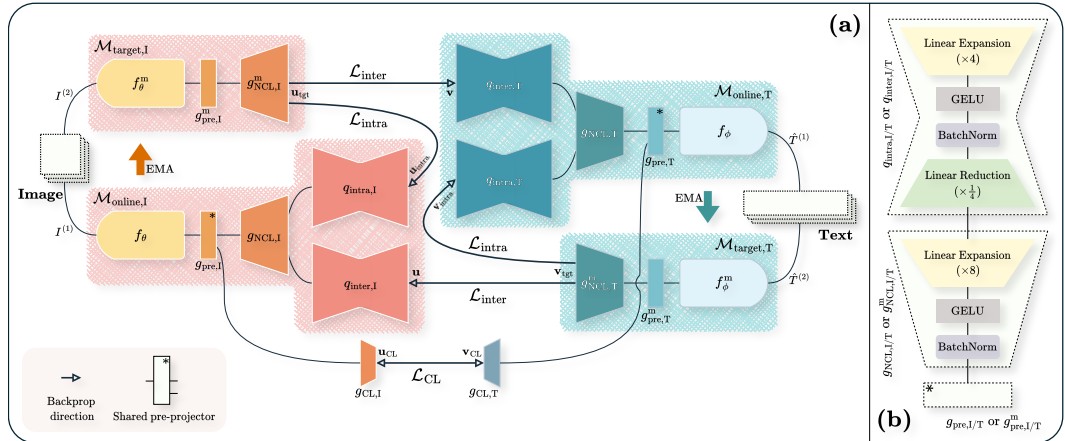

Figure 1: Overview of the proposed CLIPin framework. (a) CLIPin architecture with key modules, loss functions, and parameter update strategy. (b) Detailed structure of projectors and predictors in CLIPin.

limiting its compatibility with existing contrastive frameworks and weakening the interpretability of learned alignments.

## 3 METHOD

### 3.1 NON-CONTRASTIVE MULTIMODAL ARCHITECTURE OF CLIPIN

**Overview.** To address the limitations of CLIP in learning robust and generalizable representations, particularly its vulnerability to semantic looseness and redundancy, we propose **CLIPin**, a unified non-contrastive plug-in that can be seamlessly integrated into CLIP-style architectures to enhance cross-modal semantic alignment, inspired by momentum-based dual-branch architectures in self-supervised learning (Grill et al., 2020). Unlike the original CLIP framework, which relies exclusively on contrastive learning with negative sample pairs, CLIPin incorporates a non-contrastive pathway built on a symmetric online-target architecture for both image and text modalities. This results in parallel processing branches that facilitate both inter- and intra-modal alignment jointly. Each branch includes a modality-specific encoder, a projector, and a predictor (only on the online side). The target branch omits the predictor to introduce asymmetry and is updated via exponential moving average (EMA) of the corresponding online branch.

For each image-text pair, two random augmentations of comparable strength are independently applied to the image and text, generating distinct yet semantically consistent views for each modality. These augmented views are then processed through their modality-specific branches. CLIPin performs cross-modal alignment by treating the output of the target branch from one modality as the regression target for the online branch of the other. This supervision encourages both modalities to align within a shared semantic space, capturing cross-modal consistency without requiring negative sample pairs. Additionally, CLIPin includes an intra-modal alignment mechanism that reinforces consistency between augmented views of the same modality, further regularizing feature learning.

**Inter-modal alignment mechanism.** We now describe the architecture of CLIPin in detail, as illustrated in Fig. 1(a). For each image-text pair in a training batch, the input image is augmented by two random transformations of equal strength, producing $I^{(1)}$ and $I^{(2)} \in \mathbb{R}^{3 \times H \times W}$. The corresponding text $T$ is tokenized and augmented to obtain $\hat{T}^{(1)}$ and $\hat{T}^{(2)} \in \mathbb{R}^l$, where $l$ denotes the maximum text length.

We define four branches in total: an online and a target branch for each of the image and text modalities. These branches enable bidirectional inter-modal supervision. Specifically,

$$\mathcal{M}_{\text{online},I/T}(\cdot) = g_{I/T}\big(f_{\theta/\phi}(\cdot)\big), \quad \mathcal{M}_{\text{target},I/T}(\cdot) = g_{I/T}^{\text{m}}\big(f_{\theta/\phi}^{\text{m}}(\cdot)\big), \tag{1}$$

where $f_{\theta/\phi}$ denotes the image or text encoder, and $g_{\mathrm{I/T}}$ is the corresponding modality-specific projector, which will be elaborated in Section 3.2. The momentum versions, $f^{\mathrm{m}}_{\theta/\phi}$ and $g^{\mathrm{m}}_{\mathrm{I/T}}$, constitute the target branches. Parameters in the target branches are updated using an EMA of the online parameters:

$$
\begin{aligned}
\mathcal{M}^{0}_{\mathrm{target,I/T}} &= \mathcal{M}^{0}_{\mathrm{online,I/T}}, \\
\mathcal{M}^{t}_{\mathrm{target,I/T}} &\leftarrow \beta \cdot \mathcal{M}^{t-1}_{\mathrm{target,I/T}} + (1-\beta) \cdot \mathcal{M}^{t}_{\mathrm{online,I/T}},
\end{aligned}
\tag{2}
$$

where $t$ is the training step and $\beta$ is the momentum coefficient. $q_{\mathrm{inter,I}}$ and $q_{\mathrm{inter,T}}$ are the image and text predictors that appended to the online branches to introduce asymmetry that helps prevent collapse (Grill et al., 2020; Chen & He, 2021). The predicted features from the online branches are:

$$
\mathbf{u} = q_{\mathrm{inter,I}}\big(\mathcal{M}_{\mathrm{online,I}}(I^{(1)})\big), \quad \mathbf{v} = q_{\mathrm{inter,T}}\big(\mathcal{M}_{\mathrm{online,T}}(\hat{T}^{(1)})\big).
\tag{3}
$$

Likewise, we obtain target features:

$$
\mathbf{u}_{\mathrm{tgt}} = \mathcal{M}_{\mathrm{target,I}}(I^{(2)}), \quad \mathbf{v}_{\mathrm{tgt}} = \mathcal{M}_{\mathrm{target,T}}(\hat{T}^{(2)}).
\tag{4}
$$

Let $\mathrm{Norm}(\cdot) = \frac{\cdot}{\|\cdot\|_2}$ denote $\ell_2$ normalization, the inter-modal alignment loss $\mathcal{L}_{\mathrm{inter}}$ comprises cross-modal similarity losses in both the image-to-text (I2T) and text-to-image (T2I) directions:

$$
\begin{aligned}
\mathcal{L}_{\mathrm{inter,I2T}} &= -\mathrm{Norm}(\mathbf{u}) \cdot \mathrm{Norm}(\mathbf{v}_{\mathrm{tgt}}), \\
\mathcal{L}_{\mathrm{inter,T2I}} &= -\mathrm{Norm}(\mathbf{v}) \cdot \mathrm{Norm}(\mathbf{u}_{\mathrm{tgt}}), \\
\mathcal{L}_{\mathrm{inter}} &= \mathcal{L}_{\mathrm{inter,I2T}} + \mathcal{L}_{\mathrm{inter,T2I}}.
\end{aligned}
\tag{5}
$$

**Intra-modal alignment enhancement.** Inter-modal alignment alone may not provide sufficient optimization signals in the early stage of training, especially given the heterogeneity between image and text encoders. To address this, CLIPin incorporates an intra-modal self-alignment module that reinforces consistency within each modality. Specifically, we introduce separate predictors $q_{\mathrm{intra,I}}$ and $q_{\mathrm{intra,T}}$ for the image and text modalities, appended to the respective online branches.

The intra-modal aligned features are computed by aligning the prediction of one augmented view with the target representation of the other view within the same modality:

$$
\mathbf{u}_{\mathrm{intra}} = q_{\mathrm{intra,I}}\big(\mathcal{M}_{\mathrm{online,I}}(I^{(1)})\big), \quad \mathbf{v}_{\mathrm{intra}} = q_{\mathrm{intra,T}}\big(\mathcal{M}_{\mathrm{online,T}}(\hat{T}^{(1)})\big).
\tag{6}
$$

The corresponding intra-modal alignment loss $\mathcal{L}$intra reuses the target features from the same modality:

$$
\begin{aligned}
\mathcal{L}_{\mathrm{intra,I}} &= -\mathrm{Norm}(\mathbf{u}_{\mathrm{intra}}) \cdot \mathrm{Norm}(\mathbf{u}_{\mathrm{tgt}}), \\
\mathcal{L}_{\mathrm{intra,T}} &= -\mathrm{Norm}(\mathbf{v}_{\mathrm{intra}}) \cdot \mathrm{Norm}(\mathbf{v}_{\mathrm{tgt}}), \\
\mathcal{L}_{\mathrm{intra}} &= \mathcal{L}_{\mathrm{intra,I}} + \mathcal{L}_{\mathrm{intra,T}}.
\end{aligned}
\tag{7}
$$

## 3.2 CONTRASTIVE LEARNING FROM SHARED PRE-PROJECTORS

**Divergence between contrastive and non-contrastive learning.** Although CLIPin is a non-contrastive plug-in specifically designed to be integrated with contrastive learning in a single framework, its architectural requirements, especially the projectors, differ from those of conventional contrastive learning. While it is conceivable that a shared projector could support both paradigms, practical considerations often call for distinct designs. Empirical evidence (Chen & He, 2021; Zhou et al., 2023) suggests that non-contrastive methods typically rely on more complex projector designs, characterized by deeper architectures and higher output dimensionalities. In contrast, contrastive methods favor simpler and lower-dimensional projectors. For example, CLIP reduces encoder output to 512 dimensions via a linear layer, whereas non-contrastive approaches like SimSiam project features to 2,048 dimensions using a multi-layer perceptron (MLP). More notably, xCLIP (Zhou et al., 2023) expands the encoder output to 32,768 dimensions through a bottleneck module to achieve optimal performance.

This divergence arises from the different roles of projectors in each paradigm. In contrastive learning, the projector acts as an "information bottleneck", preserving only essential semantic content while discarding irrelevant details. This supports the alignment of semantically related image-text

pairs and the separation of unrelated ones. A high-dimensional projector may capture excessive nuisance signals, hindering generalization across modalities (Gupta et al., 2022; Ouyang et al., 2025; Huang et al., 2024; Jing et al., 2022). In contrast, non-contrastive learning does not rely on negative sample pairs, making it less sensitive to overfitting noise in high-dimensional spaces. In this case, higher-dimensional representations can be beneficial for capturing fine-grained features and improving the overall performance. Moreover, deeper projector networks help mitigate representation collapse, a known limitation of non-contrastive objectives.

**Connecting contrastive and non-contrastive learning via two shared pre-projectors.** To integrate contrastive and non-contrastive learning for enhanced representation quality, we design the projectors $(g_{\text{I/T}}, g_{\text{I/T}}^{\text{m}})$ and predictors $(q_{\text{intra,I/T}}, q_{\text{inter,I/T}})$ as bottleneck, drawing inspiration from Zhou et al. (2023) and Chen & He (2021), and decompose each projector into two components: (*i*) a shared pre-projector $(g_{\text{pre,I/T}}, g_{\text{pre,I/T}}^{\text{m}})$, and (*ii*) a CLIPin-specific sub-projector $(g_{\text{NCL,I/T}}, g_{\text{NCL,I/T}}^{\text{m}})$, as illustrated in Fig. 1(b). After this decomposition, the online and target branches for the image and text modalities are structured as:

$$\begin{aligned} \mathcal{M}_{\text{online,I/T}}(\cdot) &= g_{\text{NCL,I/T}}\big(g_{\text{pre,I/T}}\big(f_{\theta/\phi}(\cdot)\big)\big), \\ \mathcal{M}_{\text{target,I/T}}(\cdot) &= g_{\text{NCL,I/T}}^{\text{m}}\big(g_{\text{pre,I/T}}^{\text{m}}\big(f_{\theta/\phi}^{\text{m}}(\cdot)\big)\big). \end{aligned} \tag{8}$$

The shared pre-projectors $g_{\text{pre,I/T}}$ and $g_{\text{pre,I/T}}^{\text{m}}$ first map the encoder outputs $f_{\theta/\phi}$ and $f_{\theta/\phi}^{\text{m}}$ to a 1,024-dimensional space, providing a balanced intermediate representation suited to both contrastive and non-contrastive learning. The outputs are then further projected to 512 dimensions by the contrastive-specific layers $g_{\text{CL,I/T}}$ for computing the contrastive loss. Simultaneously, the outputs are expanded to 8,192 dimensions via $g_{\text{NCL,I/T}}$ and $g_{\text{NCL,I/T}}^{\text{m}}$ for computing the non-contrastive loss. The above designs accommodate both contrastive and non-contrastive learning paradigms and enables the joint optimization of their objectives, providing more informative gradients for parameter updates.

For a given sample pair, the contrastive features are computed as:

$$\mathbf{u}_{\text{CL}} = g_{\text{CL,I}}\big(g_{\text{pre,I}}\big(f_{\theta}(I^{(1)})\big)\big), \quad \mathbf{v}_{\text{CL}} = g_{\text{CL,T}}\big(g_{\text{pre,T}}\big(f_{\phi}(\hat{T}^{(1)})\big)\big), \tag{9}$$

where $g_{\text{CL,I}}$ and $g_{\text{CL,T}}$ are single-layer linear projectors for contrastive learning. Let a feature set with batch size $B$ be represented by:

$$\mathbf{U}_{\text{CL}} = \{\mathbf{u}_{\text{CL},1}, \dots, \mathbf{u}_{\text{CL},B}\}, \quad \mathbf{V}_{\text{CL}} = \{\mathbf{v}_{\text{CL},1}, \dots, \mathbf{v}_{\text{CL},B}\}, \tag{10}$$

and let $\tau$ denote the temperature coefficient, the contrastive loss $\mathcal{L}_{\text{CL}}$ is given by:

$$\begin{aligned} \mathcal{L}_{\text{CL,I2T}} &= -\frac{1}{B}\sum_{i=1}^{B} \log \frac{\exp\big(\text{Norm}(\mathbf{u}_{\text{CL},i})^{\top}\,\text{Norm}(\mathbf{v}_{\text{CL},i})/\tau\big)}{\sum_{j=1}^{B}\exp\big(\text{Norm}(\mathbf{u}_{\text{CL},i})^{\top}\,\text{Norm}(\mathbf{v}_{\text{CL},j})/\tau\big)}, \\ \mathcal{L}_{\text{CL,T2I}} &= -\frac{1}{B}\sum_{i=1}^{B} \log \frac{\exp\big(\text{Norm}(\mathbf{v}_{\text{CL},i})^{\top}\,\text{Norm}(\mathbf{u}_{\text{CL},i})/\tau\big)}{\sum_{j=1}^{B}\exp\big(\text{Norm}(\mathbf{v}_{\text{CL},i})^{\top}\,\text{Norm}(\mathbf{u}_{\text{CL},j})/\tau\big)}, \\ \mathcal{L}_{\text{CL}} &= \mathcal{L}_{\text{CL,I2T}} + \mathcal{L}_{\text{CL,T2I}}. \end{aligned} \tag{11}$$

The final total loss combines the contrastive and non-contrastive objectives as:

$$\mathcal{L} = \mathcal{L}_{\text{CL}} + \lambda_{\text{inter}} \cdot \mathcal{L}_{\text{inter}} + \lambda_{\text{intra}} \cdot \mathcal{L}_{\text{intra}}, \tag{12}$$

where $\lambda_{\text{inter}}$ and $\lambda_{\text{intra}}$ are learnable weighting coefficients.

## 4 EXPERIMENTS

### 4.1 EXPERIMENT SETTINGS

**Datasets.** For natural domain, we train on COCO (Lin et al., 2014) (82.8K images, 414.1K captions) and MUGE[1] (250.4K image-text pairs from e-commerce). Evaluation is conducted on five

---

[1]https://tianchi.aliyun.com/muge

Table 1: Classification results (AUC/mAP, %)

| | Linear probing | | | | Prompt-based OOD-ZSC | | | |
|---|---|---|---|---|---|---|---|---|
| | CLIP | xCLIP | CCSD | Ours | CLIP | xCLIP | CCSD | Ours |
| *COCO* | | | | | | | | |
| ① | 92.59/66.25 | 92.11/65.26 | 92.46/66.62 | **92.84/67.69** | 93.10/74.35 | 91.52/64.21 | 94.67/74.94 | **96.06/79.93** |
| ② | 93.15/37.87 | 92.59/35.46 | 92.85/38.01 | **93.38/38.31** | 49.74/1.43 | 49.21/1.42 | 50.27/1.45 | **51.31/1.48** |
| ③ | 90.86/13.22 | 89.33/11.90 | 89.99/14.11 | **91.61/14.54** | 96.31/**29.54** | 94.91/19.28 | 95.25/25.32 | **96.92**/24.88 |
| ④ | 87.18/41.92 | 85.95/40.34 | 86.83/42.88 | **87.43/43.43** | 91.33/76.47 | 93.81/77.74 | 93.72/81.25 | **94.90/85.47** |
| ⑤ | 92.33/39.83 | 90.81/37.58 | 91.68/39.57 | **92.55/40.39** | 93.74/47.25 | 94.08/39.92 | 94.79/47.24 | **95.57/47.69** |
| *MUGE* | | | | | | | | |
| ① | 93.21/69.58 | 93.29/69.70 | 92.58/70.48 | **93.72/71.69** | 86.89/49.82 | 90.07/60.37 | 90.09/63.74 | **92.65/67.23** |
| ② | 93.97/41.59 | 93.66/41.73 | 93.26/42.57 | **94.18/43.19** | 50.23/1.45 | 51.60/1.58 | 51.55/1.52 | **52.35/1.81** |
| ③ | 90.60/14.64 | 90.44/14.98 | **90.61**/15.00 | 90.57/**15.12** | 91.29/14.94 | 91.60/16.14 | 91.63/16.10 | **95.17/25.99** |
| ④ | 84.72/38.26 | 84.75/38.85 | 84.73/39.15 | **85.18/39.80** | 91.48/66.79 | 92.39/67.63 | 93.10/66.99 | **93.18/68.89** |
| ⑤ | 93.59/47.45 | 93.70/**49.48** | 92.93/48.19 | **93.79**/49.20 | 93.67/51.04 | **94.32**/51.01 | 93.98/51.24 | 94.17/**51.35** |
| *[Private Dataset]* | | | | | | | | |
| ⑥ | 86.76/40.87 | 86.96/41.15 | 87.23/41.52 | **88.89/41.71** | 82.60/39.98 | 82.07/40.12 | 82.94/40.50 | **84.83/44.21** |
| ⑦ | **85.24**/54.99 | 84.42/55.23 | 85.05/55.18 | 84.75/**55.34** | 86.45/54.72 | **88.04**/57.91 | 86.52/57.03 | 86.07/**59.49** |
| ⑧ | 96.64/92.92 | 95.73/**93.50** | 96.72/93.01 | **97.29**/93.39 | 86.57/87.30 | 92.09/89.61 | 90.16/90.86 | **92.99/92.80** |
| ⑨ | 74.72/50.34 | 74.06/49.60 | 75.18/50.76 | **75.34/53.31** | 67.62/41.20 | 59.27/37.82 | 69.25/44.48 | **72.89/48.88** |
| ⑩ | 94.99/88.86 | 94.24/88.15 | **95.04**/89.17 | 94.78/**89.49** | 94.56/89.27 | 95.73/90.06 | **95.85**/90.14 | 95.63/**90.75** |
| *MIMIC-CXR* | | | | | | | | |
| ⑪ | 67.80/20.96 | 67.44/21.16 | 68.27/21.18 | **69.57/22.77** | 60.06/13.49 | 53.43/11.58 | 60.25/12.96 | **60.44/14.12** |
| ⑫ | 92.90/92.70 | 92.92/93.04 | 93.00/92.91 | **93.47/93.27** | 42.08/50.50 | 42.00/48.48 | 46.25/55.14 | **58.61/58.88** |
| ⑬ | 81.57/82.30 | 82.54/82.28 | 84.27/84.75 | **88.90/90.22** | 73.32/72.16 | 70.83/75.60 | 77.59/78.92 | **80.00/83.24** |

benchmarks: ① CIFAR-10 (Krizhevsky & Hinton, 2009), ② CIFAR-100 (Krizhevsky & Hinton, 2009), ③ SUN397 (Xiao et al., 2016), ④ PASCAL VOC2007[2], and ⑤ Caltech-101 (Li et al., 2004). For medical domain, we select ophthalmology and chest radiography to evaluate the effectiveness of CLIPin. In ophthalmology, we train on a private dataset ([Private Dataset], containing 451.9K retinal image-report pairs from [Anonymous Hospital]), and evaluate on ⑥ RFMiD (Pachade et al., 2021), ⑦ ODIR[3], ⑧ REFUGE (Orlando et al., 2020), ⑨ MESSIDOR (Decencière et al., 2014), and ⑩ FIVES (Jin et al., 2022). For chest radiography, we train on MIMIC-CXR (Johnson et al., 2019) (377.1K chest X-ray image-report pairs), and evaluate on ⑪ NIH-Chest-X-ray-dataset-small[4], ⑫ Shenzhen chest X-ray set (Jaeger et al., 2014), and ⑬ Montgomery County chest X-ray set (Jaeger et al., 2014).

**Model configuration.** All models adopt ViT-B/16 (Dosovitskiy et al., 2021) as the image encoder. Models trained on COCO and MIMIC-CXR are initialized with CLIP, while those on MUGE and [Private Dataset] use CN-CLIP (Yang et al., 2022). The text encoder varies across datasets but is fixed per experiment. Input images are resized to $224 \times 224$, randomly horizontally flipped (probability $0.5$), and augmented with color jitter (strength $0.1$). The max text length $l$ is 77. We use AdamW (Loshchilov & Hutter, 2018) with a learning rate of $3 \times 10^{-5}$, warmup of 100 iterations, $\beta_1 = 0.9$, $\beta_2 = 0.98$, $\epsilon = 1 \times 10^{-6}$, and weight decay $\lambda = 0.001$. The momentum coefficient $\beta = 0.95$, temperature $\tau = 0.07$, and weighting coefficients $\lambda_{inter}$ and $\lambda_{intra}$ are initialized to 1.0, and jointly optimized with the rest of the model via standard gradient descent. The batch size $B = 256$. The training takes approximately 24 hours on single RTX 3090 GPU using automatic mixed precision and gradient checkpointing, with a memory consumption of 14 GB.

**Tasks and metrics.** We evaluate using linear probing, prompt-based out-of-distribution zero-shot classification (prompt-based OOD-ZSC), and cross-modal retrieval. Linear probing follows He et al. (2022), training a linear classifier atop frozen encoders. Prompt-based OOD-ZSC follows CLIP's standard zero-shot protocol (with prompts translated into accurate Chinese terms when evaluating models trained in Chinese) but evaluates on out-of-distribution classes, which is necessary in medical settings where test categories often appear in pre-training due to limited diagnostic vocabularies,

---

[2]http://www.pascal-network.org/challenges/VOC/voc2007/workshop/index.html

[3]https://odir2019.grand-challenge.org

[4]https://huggingface.co/datasets/Sohaibsoussi/NIH-Chest-X-ray-dataset-small

Table 2: Cross-modal retrieval results (%)

| | Text-to-image | | | Image-to-text | | |
|---|---|---|---|---|---|---|
| | R@1 | R@5 | R@10 | R@1 | R@5 | R@10 |
| CLIP | 18.03 | 37.11 | 47.85 | 27.49 | 49.77 | 57.93 |
| xCLIP | 19.04 | 37.96 | 45.20 | 25.86 | 50.94 | 60.48 |
| CCSD | 18.67 | 37.85 | 48.16 | 27.93 | 50.26 | 59.93 |
| Ours | **20.04** | **39.31** | **48.74** | **28.05** | **50.98** | **61.59** |

violating the "unseen class" assumption of standard zero-shot evaluation. This setup evaluates both generalization and modality alignment. As all datasets are multi-labeled, we report Area Under the ROC Curve (AUC) and mean Average Precision (mAP), where AUC reflects global discriminative capability and mAP captures performance on long-tailed labels. In contrast, cross-modal retrieval is evaluated on text-image downstream datasets, where candidates are ranked according to cosine similarity between their multimodal embeddings. Performance is reported using Recall@K (R@K, with K=1,5,10) for both text-to-image and image-to-text retrieval directions. The reported values are averaged over five repeated runs. Throughout the experiments, we assess statistical significance using paired $t$-tests, all improvements reported are statistically significant with $p < 0.05$. For qualitative analysis, we use multimodal Grad-CAM (Selvaraju et al., 2017) to generate heatmaps conditioned on text inputs.

## 4.2 COMPARATIVE STUDY

**Linear probing classification.** We compare the linear probing performance of CLIP (Radford et al., 2021), xCLIP (Zhou et al., 2023), CLIP with Cosmos-style (Kim et al., 2025) Cross-modality Self-Distillation (CCSD), and CLIP intergated with CLIPin (Ours). CLIP serves as the baseline; xCLIP represents a state-of-the-art fusion method that introduces a non-contrastive auxiliary loss to enhance contrastive learning; while Cosmos is a representative approach that, similar to CLIPin, incorporates cross-modality self-distillation, but differs in that it employs auxiliary distillation to encourage the contrastive learning process to attend to non-foreground regions. All models are trained from scratch under a unified training setup. As shown in Table 1 (left), CLIPin consistently improves both AUC and mAP across datasets, with notable gains on challenging categories.

When trained on the COCO dataset in natural domain, our method achieves the best results across all evaluation cases. On MUGE, CLIPin also brings significant improvements in the majority of evaluation cases. In medical domain, when trained on [Private Dataset] and MIMIC-CXR, CLIPin delivers performance gains consistently. Due to limitations of semantic looseness and redundancy, the InfoNCE loss used in CLIP often suffers from inaccurate optimization, causing semantically similar samples to be pushed apart in feature space, which undermines representation quality. xCLIP introduces non-contrastive learning to mitigate this limitation. However, since its optimization is based on batch-level distributional alignment, there exists a gap between its training objective and the contrastive learning framework, resulting in only moderate improvements in representation quality. While the CCSD offers modest improvements by adding auxiliary distillation, it remains bound to the contrastive paradigm and struggles under semantic redundancy. In contrast, due to the instance-level semantic alignment, CLIPin can be seamlessly integrated into the CLIP framework and optimized with the contrastive objective jointly, which significantly improves CLIP's representation learning performance and generalization ability.

**Prompt-based OOD-ZSC classification.** We apply prompt-based OOD-ZSC to evaluate both the quality of feature extraction capability and the alignment between visual and textual representations. Encoders of all models are fine-tuned on the same pretrained CLIP backbone to ensure effective classification performance. The results are presented in Table 1 (right).

Notably, on the PASCAL VOC2007 dataset, the model trained on COCO with our method outperforms the second-best baseline by a significant margin of +4.22 mAP. On SUN397, a challenging dataset with a large number of categories, our model trained with MUGE achieves improvements of +3.54 AUC and +9.85 mAP. In medical domain, the model trained on [Private Dataset] using our method achieves the highest performance gains on the MESSIDOR dataset for diabetic retinopathy grading. The model trained on the MIMIC-CXR dataset also achieves consistent performance

Table 3: Generalization study of CLIPin: linear probing classification results (AUC/mAP, %)

| | ALBEF (+CLIPin) | BLIP (+CLIPin) | CoCa (+CLIPin) | OTTER (+CLIPin) | SLIP (+CLIPin) |
|---|---|---|---|---|---|
| *COCO* | | | | | |
| ① | **92.31/65.12** (92.27/65.11) | **92.58/66.58** (92.28/65.91) | 89.05/54.46 (**89.83/56.87**) | 86.89/48.03 (**87.14/48.93**) | 84.70/43.92 (**84.89/44.07**) |
| ② | **92.83/35.11** (92.71/34.79) | **92.93/35.12** (92.70/34.92) | 89.60/21.33 (**90.72/25.16**) | 87.67/17.86 (**87.91/18.52**) | 84.14/14.33 (**84.57/14.44**) |
| ③ | 91.72/13.90 (**91.84/14.30**) | 92.02/14.77 (**92.13/15.46**) | 87.63/8.67 (**88.32/10.27**) | 87.99/7.15 (**88.22/7.56**) | 79.49/3.39 (**80.13/3.48**) |
| ④ | 88.02/43.52 (**88.14/44.99**) | 88.51/46.06 (**88.95/47.68**) | **86.06**/38.85 (86.03/**39.77**) | 81.94/30.62 (**82.23/31.38**) | 87.78/43.18 (**88.11/43.88**) |
| ⑤ | 91.43/36.95 (**92.51/37.82**) | 92.33/40.51 (**93.03/41.15**) | 88.50/29.64 (**90.91/34.86**) | **87.17**/24.82 (86.87/**25.72**) | 78.70/14.49 (**78.93/15.18**) |
| *MUGE* | | | | | |
| ① | **89.94**/57.31 (89.71/**57.38**) | 89.74/57.93 (**89.85/58.36**) | 86.62/47.78 (**88.17/53.84**) | 87.46/49.20 (**87.61/50.21**) | 89.91/57.67 (**90.07/59.58**) |
| ② | 90.81/28.52 (**90.92/29.06**) | 90.62/28.88 (**91.33/29.41**) | 87.30/18.54 (**88.50/22.96**) | 87.68/17.93 (**88.11/18.97**) | 90.85/29.35 (**91.22/30.70**) |
| ③ | **87.92**/8.35 (87.80/**8.64**) | 86.93/8.04 (**88.19/8.76**) | 81.03/4.38 (**82.41/5.40**) | 87.69/7.13 (**88.06/7.61**) | 87.49/8.55 (**88.23/8.89**) |
| ④ | 81.19/29.45 (**81.45/30.21**) | 81.39/30.08 (**81.92/30.87**) | 76.56/23.74 (**77.65/24.67**) | 79.83/27.22 (**79.95/27.51**) | **81.21**/28.79 (80.96/**29.78**) |
| ⑤ | 89.87/32.05 (**90.37/34.57**) | 90.25/**34.08** (**91.15**/34.05) | 86.12/25.73 (**88.40/31.54**) | **87.24**/24.89 (86.78/**25.70**) | 90.12/34.35 (**90.87/35.70**) |
| *[Private Dataset]* | | | | | |
| ⑥ | 84.01/32.33 (**85.18/35.23**) | 83.45/30.49 (**85.27/31.45**) | 79.56/24.91 (**80.29/25.50**) | 75.32/18.47 (**76.25/19.11**) | **82.06**/32.43 (79.43/**33.00**) |
| ⑦ | 82.26/48.93 (**82.41/51.92**) | 82.27/49.95 (**82.67/50.07**) | 78.73/45.93 (**79.22/46.71**) | 77.53/38.35 (**79.26/40.70**) | 80.81/49.48 (**82.38/51.36**) |
| ⑧ | 96.40/92.42 (**96.53/92.67**) | **94.47/91.57** (92.75/91.20) | **94.48**/88.36 (93.94/**88.63**) | 76.45/63.80 (**80.65/65.57**) | 96.57/90.56 (**97.04/92.80**) |
| ⑨ | **69.25**/43.36 (68.56/**43.99**) | **70.68/46.88** (68.45/46.67) | 67.35/**44.28** (**68.01**/44.03) | 60.92/39.26 (**63.16/40.54**) | 68.36/44.12 (**69.44/45.31**) |
| ⑩ | 93.09/84.98 (**93.14/85.79**) | 90.95/78.94 (**93.37/83.96**) | 91.87/81.76 (**92.96/82.59**) | 85.10/68.12 (**89.34/75.20**) | 91.96/79.83 (**92.81/83.75**) |
| *MIMIC-CXR* | | | | | |
| ⑪ | 67.74/23.67 (**72.31/25.98**) | 63.79/17.04 (**68.76/23.39**) | 69.26/23.89 (**69.45/24.27**) | 62.09/13.62 (**62.46/14.17**) | 66.41/22.19 (**69.37/23.54**) |
| ⑫ | 89.21/89.49 (**93.76/93.64**) | 86.05/84.82 (**86.62/86.29**) | 87.41/87.32 (**89.77/89.67**) | 87.41/85.68 (**88.30/86.73**) | 89.78/89.55 (**90.89/90.73**) |
| ⑬ | 83.62/85.29 (**85.78/86.18**) | 70.80/72.90 (**79.42/80.35**) | 79.09/80.34 (**81.68/83.71**) | 66.59/67.80 (**66.81/67.83**) | 81.79/83.75 (**84.48/85.86**) |

improvements across all three chest X-ray downstream datasets. These results demonstrate that CLIPin mitigates the key limitations of the original CLIP framework effectively, particularly its susceptibility to semantic looseness and redundancy. Compared to xCLIP, which improves alignment indirectly through inter-modal distribution consistency and intra-modal diversity, CLIPin enhances instance-level semantic alignment explicitly, offering clear advantages in zero-shot multimodal semantic alignment under distribution shift.

**Cross-modal retrieval.** To investigate the impact of CLIPin on cross-modal retrieval performance, we conduct text-to-image and image-to-text retrieval experiments. All models are initialized from CLIP checkpoints, trained on COCO, and evaluated on the Flickr30k Karpathy split [5]. Results are reported in Table 2.

Our method consistently improves R@K (for K = 1, 5, 10) over the CLIP baseline, xCLIP, and CCSD in both retrieval directions. This confirms that the non-contrastive losses $\mathcal{L}_{inter}$ and $\mathcal{L}_{intra}$ act as auxiliary alignment objectives that complement (rather than interfere with) the original bidirectional InfoNCE loss. By enhancing instance-level semantic alignment without altering the global contrastive optimization target, CLIPin strengthens fine-grained semantic matching while preserving the discriminative structure of the embedding space. Consequently, CLIPin not only avoids degrading retrieval performance but also yields consistent gains, providing direct evidence of improved multimodal semantic alignment.

**Generalization study of CLIPin.** To evaluate the effectiveness and plug-and-play feasibility of the proposed CLIPin, we selected several state-of-the-art methods known for enhancing the robustness of contrastive learning: ALBEF (Li et al., 2021a), BLIP (Li et al., 2022b), CoCa (Yu et al., 2022), OTTER (Wu et al., 2022), and SLIP (Mu et al., 2022). ALBEF improves vision-language pretraining via momentum-based feature alignment and contrastive objectives; BLIP leverages bootstrapped captions and weakened supervision signals to enrich visual-language alignment; CoCa combines contrastive and generative learning in a unified multimodal framework; OTTER distills soft image-text correspondences derived from optimal-transport matching to enhance contrastive learning; SLIP strengthens visual representations by pairing CLIP-style language supervision with self-supervised image objectives. All models are trained from scratch to ensure a fair comparison. Critically, we retain all original auxiliary losses, training schedules, and architectural components unchanged: no loss weights, hyperparameters, labels, pretext tasks, or supervision signals are modified or added. The only addition is the CLIPin non-contrastive branch, which operates in paral-

---

[5]https://www.kaggle.com/datasets/shtvkumar/karpathy-splits?select=dataset_flickr30k.json

Table 4: Ablation study on linear probing classification results (AUC/mAP, %)

| | | | | | |
|---|---|---|---|---|---|
| Contrastive Learning | ✓ | ✓ | ✓ | ✓ | ✓ |
| Inter-modal Alignment | | ✓ | | ✓ | ✓ |
| Intra-modal Alignment | | | ✓ | ✓ | ✓ |
| Shared Pre-projectors | | | | | ✓ |
| PASCAL VOC2007 | 87.18/41.92 | 87.23/41.91 | 87.17/42.04 | 87.03/42.57 | **87.43/43.43** |
| RFMiD | 86.76/40.87 | 86.44/39.77 | 00.00/00.00 | 88.62/41.04 | **88.89/41.71** |

lel without replacing or altering any existing module. We integrated CLIPin into their contrastive learning modules and compared the linear probing classification performance before and after this integration, as shown in Table 3, to demonstrate that CLIPin can further enhance these advanced frameworks.

The integration of CLIPin yields measurable improvements in both AUC and mAP consistently, demonstrating its broad applicability and plug-in effectiveness. On COCO, CLIPin contributes most significantly to CoCa, boosting mAP by +2.41 on CIFAR-10 and +5.22 on Caltech-101. Although ALBEF and BLIP already employ momentum-based distillation mechanisms, they still benefit from CLIPin with consistent gains. For instance, +1.62 mAP in BLIP on PASCAL VOC2007 and +0.87 mAP in ALBEF on Caltech-101. When trained on MUGE, CoCa again gains notably, with improvements of +6.06 mAP on CIFAR-10 and +5.81 mAP on Caltech-101, while ALBEF shows up to +2.52 mAP. On [Private Dataset], CLIPin continues to provide robust enhancements. For instance, AUC increases by +1.17 for ALBEF and +1.82 for BLIP on RFMiD, while mAP increases by +2.42 for BLIP and +7.08 for OTTER on FIVES. Even in already high-performing cases such as REFUGE, CLIPin maintains or improves performance slightly. On MIMIC-CXR, CLIPin delivers consistent performance improvements, with particularly notable gains of +8.62 AUC and +7.45 mAP for BLIP on the Montgomery County chest X-ray dataset. The results indicate that although existing methods employ complex and effective constraints to improve representation quality, they still lack mechanisms that enhance contrastive representation learning through non-contrastive semantic alignment. CLIPin addresses this gap and provides consistent improvements when incorporated into these frameworks.

### 4.3 ABLATION STUDY

To assess the contribution of each component in CLIPin, we perform ablation studies in Table 4 using the COCO and [Private Dataset] as training datasets, evaluating linear probing classification performance on two downstream benchmarks: PASCAL VOC2007 and RFMiD. Starting from a baseline CLIP model, we incorporate the three key modules of CLIPin: inter-modal alignment, intra-modal alignment, and pre-projector sharing sequentially, and analyze the impact of each.

The results reveal several noteworthy trends. First, incorporating inter-modal alignment alone provides marginal improvements, and in some cases even slightly degrades the performance. This suggests that isolated cross-modal alignment, especially when implemented via a momentum-based target encoder, may introduce instability or convergence difficulties in the early training stage. The lack of anchoring in the unimodal space makes it harder for the model to form robust semantic correspondences across modalities. Introducing intra-modal alignment alleviates these issues, leading to clearer gains across tasks. Finally, adding the two shared pre-projectors further boosts the performance, confirming that unifying parts of the architecture across learning paradigms does not interfere with, and may even synergize dual training objectives. This validates the effectiveness of the plug-in design of CLIPin, demonstrating that its benefits not only rely on isolated modules but also emerge from their joint interaction.

### 4.4 MULTIMODAL GRAD-CAM VISUALIZATION

To illustrate how CLIPin enhances the interpretability of features learned by CLIP more intuitively, we adopt multimodal Grad-CAM for visualization. In natural domain, the model is trained on COCO and evaluated on PASCAL VOC2007. In medical domain, the model is trained on [Private Dataset] and evaluated on FGADR (Zhou et al., 2020), which includes pixel-level lesion annotations to enable a precise assessment of whether the activated regions are correspond to the pathological

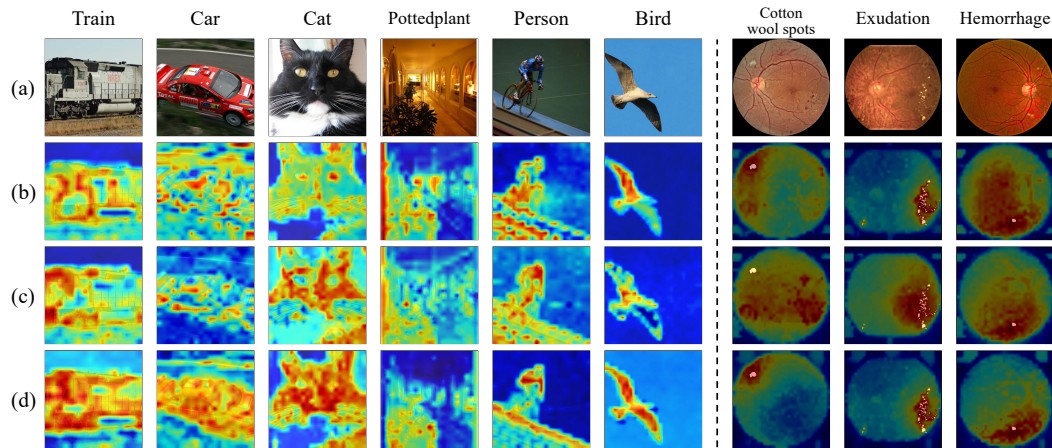

Figure 2: Multimodal Grad-CAM visualization. Each column shows the activation map for a given category text applied to the corresponding image. (a) Reference images. (b-d) Grad-CAM activation maps generated from models trained with CLIP, xCLIP, and CLIP with CLIPin, respectively. For retinal images, the activation maps are overlaid with pixel-level ground truth.

areas. As shown in Fig. 2, we compare Grad-CAM activation maps generated from models trained with CLIP, xCLIP, and CLIP with CLIPin.

In natural domain (column "Train"-"Bird"), CLIP with CLIPin yields denser and more spatially continuous activations that follow the shape and boundaries of target objects, while suppressing irrelevant background signals. In medical domain (column "Cotton wool spots"-"Hemorrhage"), CLIPin improves the alignment of text to visual attention significantly, enabling more accurate localization of lesion areas in terms of appearance, position, and spatial extent, with better correspondence to expert-annotated ground truth. The improved localization and semantic focus suggest that CLIP with CLIPin is better equipped to capture domain-specific visual cues, which is due to the additional instance-level supervision introduced by the non-contrastive component. These qualitative results further support our quantitative findings: integrating CLIPin into CLIP not only boosts performance metrics but also enhances the interpretability, semantic consistency, and zero-shot generalization of the learned representations.

## 5 CONCLUSION

In this work, we propose CLIPin, a unified non-contrastive plug-in designed to enhance multimodal semantic alignment. CLIPin can be seamlessly integrated into existing contrastive learning pipelines, functioning as a plug-and-play module that improves representation quality, generalization, and cross-modal alignment. By introducing additional non-contrastive pathways, CLIPin addresses the key limitations of CLIP-style models, such as semantic looseness and redundancy. Extensive experiments demonstrate that CLIPin outperforms prior methods and delivers robust performance gains across diverse architectures consistently (even under practical resource constraints such as medium-scale data and limited batch sizes), highlighting its effectiveness in realistic settings, with scaling to larger regimes being a key avenue for future work. Although CLIPin is implemented with a cyclic and modality-symmetric design that can be naturally extended to more than two modalities, this work focuses on the image-text setting due to practical constraints. Extending CLIPin beyond the bimodal case remains a promising direction for future research. We also plan to further explore the synergy between contrastive and non-contrastive paradigms, including improving robustness to data augmentations and scaling to larger multimodal corpora.

## 6 REPRODUCIBILITY STATEMENT

We have taken several steps to ensure the reproducibility of our results. The anonymous implementation of the proposed algorithm is included as part of the supplementary material. Detailed descriptions of the experimental settings and hyperparameters are provided in the Experiments section. In addition, the data collection and preprocessing procedures for the [Private Dataset] are fully documented in the Appendix.

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

# A APPENDIX

## A.1 QUANTITATIVE EVIDENCE FOR A UNIFIED VIEW OF LOOSE ALIGNMENT AND SEMANTIC REDUNDANCY

Table 5: Results of NSMR and Alignment Entropy

|  | NSMR (%) | | Alignment Entropy | |
| --- | --- | --- | --- | --- |
|  | CLIP | Ours | CLIP | Ours |
| Natural | 28.90 | **22.38** | 4.84 | **4.72** |
| Medical | 31.36 | **28.60** | 5.96 | **5.67** |

Because our pretraining data lack dense semantic annotations necessary to label false negatives directly, we evaluate on downstream datasets with multi-label ground truth. Specifically, we compute two metrics: Negative Sample Misclassification Rate (NSMR) and Alignment Entropy, on random batches ($B = 512$) drawn from PASCAL VOC2007[6] (natural domain, trained on COCO (Lin et al., 2014)) and NIH-ChestX-ray-small[7] (medical domain, trained on MIMIC-CXR (Johnson et al., 2019)).

Results are summarized in Table 5. Both domains exhibit high NSMR, confirming that semantically overlapping samples frequently appear as negatives. CLIPin consistently reduces NSMR, demonstrating its effectiveness in mitigating false negative interference. Moreover, CLIPin achieves lower alignment entropy across both natural and medical settings. This reduction suggests that the non-contrastive intra-modal pathway stabilizes representation learning by sidestepping the adverse effects of false negatives. Together, these findings support our central claim: both loose alignment and low textual diversity cause systematic violations of the InfoNCE (Oord et al., 2018) negative-pair assumption, and CLIPin provides a unified and principled solution that addresses this shared failure mode.

## A.2 DATA COLLECTION AND CURATION

In this work, we position our method within the general medical domain. For empirical validation, we focus on ophthalmology, where we constructed a large-scale dataset of retinal images paired with diagnostic reports. The dataset [Private Dataset] was collected through a large-scale telemedicine initiative involving 172 hospitals across mainland China between 2012 and 2020. Patient submissions included demographic information, clinical complaints, medical histories, ophthalmic measurements (e.g., visual acuity and intraocular pressure), and retinal images. Experienced ophthalmologists subsequently reviewed these cases and provided diagnostic reports.

To the best of our knowledge, no comparable public resource of retinal image-text pairs currently exists. Due to strict privacy regulations and institutional policies, the dataset cannot be publicly released, despite thorough de-identification. While our approach is modality-agnostic and can readily extend to public datasets in other medical domains, this paper focuses on the ophthalmology case study, and we consider broader multi-domain validation an important direction for future work.

**Data acquisition.** In total, we obtained over 900K ophthalmic images from more than 400K patient visits. A multi-stage curation pipeline was implemented to ensure data quality and consistency, combining automated classification, report-based filtering, and redundancy reduction. After this process, we retained approximately 451.9K high-quality retinal images, each paired with a corresponding diagnostic text.

**Report structure.** Each diagnostic report contains two major components: (*i*) findings, which describe observable fundus features and abnormalities; and (*ii*) impression, summarizing the case into a preliminary diagnosis and clinical recommendation.

---

[6]http://www.pascal-network.org/challenges/VOC/voc2007/workshop/index.html
[7]https://huggingface.co/datasets/Sohaibsoussi/NIH-Chest-X-ray-dataset-small

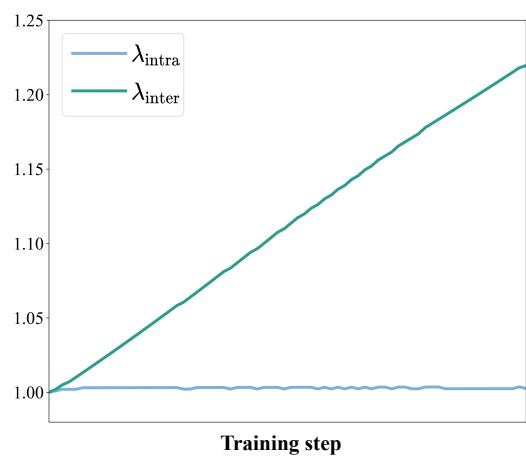

Figure 3: Training dynamics of $\lambda_{\text{inter}}$ and $\lambda_{\text{intra}}$.

Table 6: Ablation study on $\lambda_{\text{inter}}$ and $\lambda_{\text{intra}}$

| $\lambda_{\text{inter}}/\lambda_{\text{intra}}$ | 1/0 (fixed) | 0/1 (fixed) | 0.5/1 (fixed) | 1/0.5 (fixed) |
|---|---|---|---|---|
| Linear Probing mAP (%) | 87.23/42.20 | 86.98/42.01 | 87.00/42.72 | 87.19/42.43 |
| $\lambda_{\text{inter}}/\lambda_{\text{intra}}$ | 1/1 (fixed) | 1/2 (fixed) | 2/1 (fixed) | Ours (learnable) |
| Linear Probing mAP (%) | 87.31/42.86 | **87.44**/43.39 | 87.25/42.97 | 87.43/**43.43** |

This pipeline yielded a modality-consistent dataset suitable for model training and evaluation, while ensuring compliance with both ethical and privacy requirements.

### A.3 DETAILS OF DATA AUGMENTATION

Our full image augmentation pipeline is: Resize ($224 \times 224$) $\rightarrow$ RandomResizedCrop (scale=[0.8, 1.0], ratio=1.0) $\rightarrow$ RandomHorizontalFlip (p=0.5) $\rightarrow$ ColorJitter (strength=0.1) $\rightarrow$ ToRGB $\rightarrow$ ToTensor $\rightarrow$ Normalize, applied consistently across all models.

For text input, we do not apply synthetic augmentations (e.g., paraphrasing). On datasets with multiple captions per image (e.g., COCO), we use a single pre-selected caption per image in our main experiments. Although we have explored sampling different captions per iteration as natural text views, the performance is slightly degraded, which is likely due to semantic divergence among captions (e.g., focusing on different image aspects). Thus, text augmentation is disabled in all reported results.

Nevertheless, since CLIPin is designed to support text augmentation as a modular component, we retain this capability in the architecture and plan to revisit it in future work with more robust augmentation strategies.

### A.4 TRAINING DYNAMICS OF INTER- AND INTRA-MODAL ALIGNMENT WEIGHTS

We collect the values of weights $\lambda_{\text{inter}}$ and $\lambda_{\text{intra}}$ throughout training and plot their trajectories over training steps, as shown in Fig. 3. The curves cover the training period up to the point where the model reaches peak evaluation performance, after which overfitting begins and the dynamics of both weights remain stable. It can be observed that $\lambda_{\text{inter}}$ increases steadily, while $\lambda_{\text{intra}}$ remains relatively stable. This suggests that, as training progresses, inter-modal alignment gradually becomes the primary driver of representation learning, surpassing contrastive learning and intra-modal alignment in contributing to performance gains.

Table 7: Ablation study on the output dimensionality of shared pre-projectors (AUC/mAP, %)

|  | 512 | 1024 | 2048 |
|---|---|---|---|
| ViT-B/16, PASCAL VOC2007 | 87.22/43.16 | **87.43/43.43** | 87.22/41.84 |
| ViT-B/16, NIH-Chest-X-ray-dataset-small | 67.39/22.45 | **67.80/22.77** | 66.48/19.12 |
| ViT-L/14, PASCAL VOC2007 | 88.46/46.09 | **88.66/47.07** | 86.32/43.20 |

Table 8: Ablation study on varying batch sizes $B$ (AUC/mAP, %)

| $B$ | CLIP | Ours |
|---|---|---|
| 128 | 86.65/41.36 | **87.49/43.33** |
| 256 | 87.18/41.92 | **87.43/43.43** |
| 512 | 87.08/42.34 | **87.89/43.70** |

**Learnable vs. fixed loss weights.** To assess the benefit of learnable weighting, we compare it against fixed weight configurations via a small grid search. Specifically, we evaluate linear probing performance on PASCAL VOC2007 (with the model pretrained on COCO). As shown in Table 6, the learned weights either match or slightly outperform the best fixed configuration identified through grid search. While a well-tuned fixed weighting can achieve comparable results, it requires explicit manual tuning across datasets and tasks, which our learnable approach avoids.

Learnable weighting allows automatic adaptation to varying levels of semantic noise and label diversity. We emphasize that this mechanism is not intended as a theoretical guarantee, but rather as a practical, data-driven strategy to dynamically balance inter- and intra-modal supervision during training.

## A.5 ADDITIONAL ABLATION STUDIES

**Ablation of output dimensionality of the shared pre-projectors.** To evaluate the impact of the output dimensionality of the shared pre-projectors, we conduct an ablation study by varying the output dimensions across a range of values (512, 1024, and 2048) while keeping all other hyperparameters fixed. Meanwhile, to assess whether the optimal pre-projector dimension generalizes across datasets and backbone architectures, we train models on COCO and MIMIC-CXR, and evaluate them separately on PASCAL VOC2007 and NIH-Chest-X-ray-dataset-small using linear probing classification. For the former (COCO-PASCAL VOC2007), we additionally train models with the ViT-L/14 (Dosovitskiy et al., 2021) backbone to examine cross-architecture consistency.

As summarized in Table 7, we observe that the 1,024-dimensional pre-projector generalizes well across datasets, domains, and backbone architectures. The configuration used in our main experiments corresponds to the optimal output dimensionality. Increasing or decreasing the dimension beyond this setting leads to performance degradation, suggesting that our chosen configuration strikes a good balance between representation quality and optimization stability.

**Ablation of varying batch sizes.** To investigate whether integrating CLIPin into CLIP (Radford et al., 2021) reduces the reliance on large batch sizes, we compare the performance of the original CLIP and CLIP with CLIPin under varying batch sizes $B$ of 128, 256, and 512. We use COCO as the training dataset and evaluate the linear probing performance on PASCAL VOC2007. Results are reported in Table 8. We observe that across different batch size settings, CLIPin consistently improves the quality of the learned representations.

**Ablation of data augmentation.** To evaluate the impact of data augmentation on model performance, we disable either image augmentation (i.e., setting $I^{(1)} = I^{(2)}$) or text augmentation (i.e., setting $\hat{T}^{(1)} = \hat{T}^{(2)}$) selectively, and compare the results against the baseline where both image and text augmentation are disabled. Given that the COCO dataset provides rich textual augmentation (i.e., multiple captions per image), we use it as the training dataset and evaluate linear probing and prompt-based OOD-ZSC classification performance on PASCAL VOC2007. Results are shown in Table 9. We observe that enabling image augmentation improves both visual representation learning

Table 9: Ablation study on image and text augmentation (AUC/mAP, %)

| | | ✓ | | ✓ |
|---|---|---|---|---|
| Image augmentation | | ✓ | | ✓ |
| Text augmentation | | | ✓ | ✓ |
| Linear probing | 87.42/42.67 | 87.43/**43.43** | 87.48/42.71 | **87.63**/43.22 |
| Prompt-based OOD-ZSC | 79.50/29.03 | **82.13/32.20** | 81.92/30.74 | 81.01/31.98 |

Table 10: Ablation study on inter- and intra-modal alignments

| | | | | CLIP with CLIPin |
|---|---|---|---|---|
| Inter-modal Alignment | ✓ | | ✓ | |
| Intra-modal Alignment | | ✓ | ✓ | |
| AUC/mAP, % | 85.43/35.85 | 75.08/21.22 | 87.42/41.79 | **87.43/43.43** |

and multimodal semantic alignment substantially. In contrast, the benefits of text augmentation are relatively limited and, in some cases, may even diminish the gains introduced by image augmentation. Therefore, in our main experiments, we only enable image augmentation to maximize overall performance.

**Ablation of modal alignments.** To further investigate the respective contributions of non-contrastive inter- and intra-modal alignment, we train CLIPin independently without CLIP loss and conduct ablation studies by disabling each alignment component selectively. Specifically, we compare three variants: using only inter-modal alignment, using only intra-modal alignment, and using both. We use COCO as the training dataset and evaluate the linear probing performance on PASCAL VOC2007. The ablation results of CLIPin with different alignment configurations are reported in Table 10, with CLIP with CLIPin included as a reference. The results reveal that using only inter-modal alignment leads to poor representation quality, consistent with our earlier analysis that multimodal heterogeneous encoders are prone to collapse when optimized without intra-modal regularization. On the other hand, using only intra-modal alignment also results in poor performance due to the absence of multimodal supervision, which makes the encoders overly sensitive to data augmentations. Combining intra- and inter-modal alignment mitigates these issues significantly, yielding results that are close to those obtained with contrastive learning.

Combining insights from Fig. 3 and Table 10, we conclude that CLIPin and contrastive learning complement each other. In the early stages of training, the clear optimization signals from contrastive learning and intra-modal alignment help stabilize and guide the inter-modal non-contrastive objective. As training progresses, inter-modal alignment gradually takes over as the primary driver of representation learning and semantic alignment, compensating for the limitations of contrastive modeling effectively.

**Ablation of predictors.** In Section 3.1, we briefly attribute collapse prevention to "asymmetry". Here, we provide a more precise explanation.

While BYOL (Grill et al., 2020) introduced architectural asymmetry via a predictor, the theoretical justification for its role was formalized in SimSiam (Chen & He, 2021), which we follow. In a symmetric architecture without negative pairs, minimizing the distance between two views naturally leads to a trivial constant solution. Introducing a predictor $h$ (corresponding to our $q_{\text{inter}}$ and $q_{\text{intra}}$) on the online branch, together with stop-gradient on the target branch, fundamentally alters the optimization dynamics.

As analyzed in SimSiam, this design behaves similarly to an Expectation-Maximization (EM) procedure: (*i*) The target encoder acts as a stable estimator of the representation (analogous to the E-step). (*ii*) The online branch, via the predictor, updates its parameters to match this estimate (analogous to the M-step).

The predictor $h$ provides the necessary transformation capacity that allows the online representation to fit the target representation's distribution, rather than collapsing to a constant. Removing $h$ forces the encoder to reduce the objective by degenerating toward a uniform constant mapping, thereby triggering collapse.

Table 11: Ablation study on w/o and w predictors

|  | Collapse | Linear Probing mAP (%) |
|---|---|---|
| w/o predictors | Yes | 10.05 |
| w/ predictors | No | 43.43 |

To empirically validate the role of the predictors within our CLIPin framework, we conduct an ablation study in which the predictors are removed while keeping all other components unchanged. Models are trained on COCO and evaluated on PASCAL VOC2007. As shown in Table 11, removing the predictors leads to representation collapse: evidenced by a sharp drop in linear probing mAP to 10.05%, close to random performance given the dataset's class distribution. This demonstrates that the predictors are not merely architectural asymmetries but are essential for enabling the online encoder to learn meaningful representations by predicting the target features.

### A.6 DETAILED RESULTS BY CATEGORY ON DOWNSTREAM BENCHMARKS

To demonstrate the improvement in representation learning quality brought by CLIPin more concretely, we compare CLIP, xCLIP (Zhou et al., 2023), and CLIP with CLIPin (Ours) on a per-category evaluation using both linear probing and prompt-based OOD-ZSC. The models are trained on COCO and evaluated on PASCAL VOC2007. Evaluation metrics include AUC and Average Precision (AP). The detailed results are illustrated in Table 12. Categories A-T correspond to "aeroplane", "bicycle", "bird", "boat", "bottle", "bus", "car", "cat", "chair", "cow", "diningtable", "dog", "horse", "motorbike", "person", "pottedplant", "sheep", "sofa", "train", and "tvmonitor". The results show that the performance gains introduced by CLIPin are consistent across the vast majority of categories, rather than being concentrated in a few outliers. Moreover, CLIP with CLIPin generally outperforms xCLIP. These findings provide further evidence of the robustness and effectiveness of CLIPin in enhancing representation quality.

Table 12: Per-category results on downstream benchmarks (AUC/AP, %)

|  | Linear probing | | | Prompt-based OOD-ZSC | | |
|---|---|---|---|---|---|---|
|  | CLIP | xCLIP | Ours | CLIP | xCLIP | Ours |
| A | 93.49/61.19 | 92.86/61.23 | **94.08/67.19** | 82.56/11.66 | 85.84/**29.48** | **87.89**/18.56 |
| B | **86.54/33.80** | 82.53/30.37 | 85.79/33.68 | 82.12/**28.06** | 75.87/15.84 | **83.38**/25.91 |
| C | 84.46/33.79 | 83.70/33.11 | **85.17/33.87** | **82.43**/27.92 | 81.29/30.89 | 82.08/**31.36** |
| D | 91.00/51.41 | 88.94/**53.47** | **92.13**/53.16 | 90.15/37.74 | 90.49/**46.52** | **91.03**/40.48 |
| E | 80.20/15.63 | **80.40/15.68** | 78.18/13.39 | 61.29/6.35 | 57.54/6.73 | **74.40/9.35** |
| F | 87.96/31.10 | 88.59/28.21 | **89.24/32.71** | 85.17/33.75 | 84.49/32.14 | **87.85/35.65** |
| G | 86.38/56.51 | 85.47/55.29 | **86.56/57.40** | 83.86/49.94 | 81.58/43.96 | **84.76/55.05** |
| H | **86.95/36.75** | 85.74/35.75 | 86.36/35.62 | 83.80/33.27 | 84.52/31.93 | **85.19/33.78** |
| I | 84.32/46.16 | 84.34/44.66 | **85.73/46.33** | 79.65/29.46 | 82.88/40.73 | **83.52/44.19** |
| J | 88.84/22.65 | 84.97/20.47 | **89.09/23.57** | 84.72/14.00 | 82.65/15.40 | **88.53/21.52** |
| K | 89.89/34.57 | 88.33/31.71 | **90.59/39.24** | 77.87/21.41 | 81.14/14.79 | **85.03/37.00** |
| L | 80.14/31.12 | 81.30/30.10 | **82.93/32.49** | 78.47/29.48 | 78.88/29.33 | **81.27/30.85** |
| M | **91.95/61.96** | 90.42/58.84 | 91.81/61.45 | 87.29/35.92 | **88.90/42.10** | 88.42/36.70 |
| N | 86.62/40.33 | 85.34/37.41 | **87.19/42.52** | 81.15/24.46 | 78.45/23.30 | **87.35/36.18** |
| O | 83.55/81.56 | 82.74/81.52 | **83.61/82.29** | 73.07/71.56 | 72.49/69.62 | **73.51/72.33** |
| P | 83.53/28.95 | 80.41/25.49 | **84.54/29.50** | **71.82/15.38** | 59.06/10.76 | 51.91/9.33 |
| Q | 89.15/**40.98** | **90.18**/31.91 | 89.56/39.37 | **89.69**/30.39 | 88.37/27.30 | 87.52/**30.70** |
| R | 88.00/**36.98** | 85.37/28.01 | **88.14**/36.52 | 83.22/**33.24** | 83.44/28.98 | **86.74**/31.77 |
| S | 92.87/52.36 | 91.95/50.78 | **93.45/57.55** | **88.40**/38.01 | 87.48/38.21 | 87.89/**46.98** |
| T | 87.19/39.25 | 87.55/36.98 | **88.76/42.63** | 71.28/15.61 | 75.82/16.77 | **86.41/39.59** |

