# OpenReview forum: "CLIPin: A Non-contrastive Plug-in to CLIP for Multimodal Semantic Alignment"
_ICLR.cc/2026/Conference — ICLR 2026 Conference Withdrawn Submission_

### Official Review · Reviewer_1rpH · 2025-10-25

**Soundness:** 2
**Presentation:** 2
**Contribution:** 2
**Rating:** 4
**Confidence:** 4

**Summary:**

The paper proposes a pluggable non-contrastive branch “CLIPin” that introduces an online/target branch (EMA) and prediction head without modifying the original CLIP-style dual encoder architecture. It performs dual augmentation on images and text respectively, jointly minimizing: Inter-modal regression consistency loss and intra-modal consistency loss are jointly optimized with the standard InfoNCE contrastive loss. To accommodate both contrastive and non-contrastive learning, the authors decompose the projection head into a “shared pre-projector (shared across both paradigms) + distinct sub-projectors for each,” enabling simultaneous computation of 512-dimensional contrastive features and 8192-dimensional non-contrastive features during training. Experiments demonstrate superior performance across natural domains, and medical domains. Visualizations and ablation studies are provided. Authors report AUC/mAP improvements across multiple benchmarks and demonstrate comparable or beneficial results when integrated as plugins within various frameworks.

**Strengths:**

1. We provide the complete forward and target branches, EMA updates, loss formulations, and training combinations, and explicitly illustrate the data flow and module interfaces in Figure 1 for implementation and verification.

2. We progressively incorporate cross-modal/intra-modal alignment and shared pre-projection, while also examining pre-projection dimensions, batch sizes, image/text augmentation activation, and removing specific alignment components. This yields a comprehensive evidence chain.

**Weaknesses:**

1. The claimed effectiveness in the medical domain relies on pretraining with over 450K private retina image–report pairs, which are not publicly available even in anonymized form; this severely limits external reproducibility and independent verification of the results.
2. The reported “universal improvement” is not consistent across frameworks—for example, in Table 2 the BLIP model on REFUGE drops from 94.47 to 92.75 AUC, indicating that the gain depends on specific architectures, datasets, and hyperparameters.
3. Despite emphasizing multimodal semantic alignment, the paper evaluates mainly on classification metrics (AUC, mAP) and omits retrieval-based measures like Recall@K, leaving the alignment improvement only indirectly evidenced.
4. The authors disable text augmentation in main experiments due to marginal or negative effects, which contradicts the method’s core idea of constructing two consistent views for *each* modality and may weaken the non-contrastive text branch.
5. The optimization and stability of the learnable weights λ₍inter₎ and λ₍intra₎ are under-specified—Appendix Fig. 3 shows λ₍inter₎ increasing while λ₍intra₎ stays flat, but the paper does not clarify optimizer choice, constraints, or regularization, raising concerns of dominance or overfitting.

**Questions:**

1. How are the learnable weights ( \lambda_{\text{inter}} ) and ( \lambda_{\text{intra}} ) optimized in practice (optimizer, parameterization, constraints/regularization), and how stable are they across datasets and scales (cf. Fig. A.1)?
2. Since text augmentation is largely disabled in the main experiments (Table 6), how do you justify the non-contrastive text branch learning two “independent” views—does this weaken the claimed bidirectional alignment?
3. Can you report retrieval metrics (e.g., Recall@K for image→text/text→image) to directly evidence cross-modal alignment improvements, in addition to AUC/mAP classification?
4. In Table 2/8, some frameworks show degradations after adding CLIPin (e.g., BLIP on REFUGE). What factors (projector dimensions, EMA momentum, batch size, λ-schedules) drive these negative cases?
5. How sensitive is CLIPin to the choice of projector dimensions beyond the single VOC2007 study (Table 4)? Does the 1,024-dim pre-projector generalize across domains and backbones?
6. For medical-domain claims, given the private training corpus (Appendix A.1), can you provide a public surrogate (e.g., MIMIC-CXR, ROCO) to let the community verify your observed gains?
7. What collapse-avoidance ablations have you tried beyond adding predictors and EMA (e.g., stop-grad placements, predictor depth), especially for heterogeneous encoders where “inter-only” failed badly (Table 7)?
8. Your OOD-ZSC setting relies on prompt engineering (Section 4.1). How robust are the conclusions to prompt templates and label wording (especially for Chinese vs. English prompts)?
9. Training on COCO/MUGE with ViT-B/16 (Section 4.1) is relatively small-scale for CLIP-style methods. Do the gains persist or change when scaling data/backbones (e.g., ViT-L/14, larger noisy corpora)?
10. Could you clarify compute fairness: are all baselines retrained “from scratch” under identical augmentations, batch sizes, projector dims, and optimizer schedules—especially where weighted attention or auxiliary heads differ?

**If you address my concerns, I will consider raising my score.**

---

> ### Author Response · Authors · 2025-11-24
> **1. On the concern of dataset reproducibility and private pretraining corpus.**
>
> > **Weakness 1:** The claimed effectiveness in the medical domain relies on pretraining with over 450K private retina image-report pairs, which are not publicly available even in anonymized form; this severely limits external reproducibility and independent verification of the results.
> >
> > **Question 6:** For medical-domain claims, given the private training corpus (Appendix A.1), can you provide a public surrogate (e.g., MIMIC-CXR, ROCO) to let the community verify your observed gains?
>
> We thank the reviewer for this insightful comment. We fully agree that reproducibility and independent verification are essential for scientific progress. To address this important concern, we have supplemented our study with new experiments conducted exclusively on publicly available medical image benchmarks, demonstrating that the effectiveness of our method does not depend on the private pretraining corpus.
>
> Specifically, we retrained our entire framework using only the public MIMIC-CXR [1] dataset and evaluated the learned representations on multiple widely adopted chest X-ray benchmarks, including NIH-ChestX-ray-dataset-small [2], Shenzhen Chest X-ray Set [3], and Montgomery County X-ray Set [3]. The performance trends remain consistent with those reported in the main paper, confirming that our approach achieves competitive or superior results compared to strong baselines in both linear probing and zero-shot settings.
>
> The results (reported in AUC/mAP, %) is summarized below:
>
> **Evaluation on NIH-Chest-X-ray-dataset-small:**
>
> | | CLIP [6] | xCLIP [8] | Ours |
> |-----|-----|-----|-----|
> | Linear probing | 67.80/20.96 | 67.44/21.16 | 69.57/22.77 |
> | Prompt-based OOD-ZSC | 60.06/13.49 | 53.43/11.58 | 60.44/14.12 |
>
> **Evaluation on Shenzhen Chest X-ray Set:**
>
> | | CLIP [6] | xCLIP [8] | Ours |
> |-----|-----|-----|-----|
> | Linear probing | 92.90/92.70 | 92.92/93.04 | 93.47/93.27 |
> | Prompt-based OOD-ZSC | 42.08/50.50 | 42.00/48.48 | 58.61/58.88 |
>
> **Evaluation on Montgomery County X-ray Set:**
>
> | | CLIP [6] | xCLIP [8] | Ours |
> |-----|-----|-----|-----|
> | Linear probing | 81.57/82.30 | 82.54/82.28 | 88.90/90.22 |
> | Prompt-based OOD-ZSC | 73.32/72.16 | 70.83/75.60 | 80.00/83.24 |
>
> **Generalization study results (linear probing):**
>
> | ALBEF [9] (+CLIPin) | BLIP [10] (+CLIPin) | CoCa [11] (+CLIPin) |
> |-----|-----|-----|
> | 67.74/23.67 (72.31/25.98) | 63.79/17.04 (68.76/23.39) | 69.26/23.89 (69.45/24.27) |
>
> Full experimental details and expanded analysis will be included in the revised paper.

---

> ### Author Response · Authors · 2025-11-24
> **2. On the concern of performance fluctuations in certain frameworks.**
>
> > **Weakness 2:** The reported “universal improvement” is not consistent across frameworks—for example, in Table 2 the BLIP model on REFUGE drops from 94.47 to 92.75 AUC, indicating that the gain depends on specific architectures, datasets, and hyperparameters.
> >
> > **Question 4:** In Table 2/8, some frameworks show degradations after adding CLIPin (e.g., BLIP on REFUGE). What factors (projector dimensions, EMA momentum, batch size, \lambda-schedules) drive these negative cases?
>
> We appreciate the reviewer for this observation. Indeed, a slight drop (94.47 → 92.75 AUC) is observed for BLIP [7] on the REFUGE [12] dataset, while other frameworks and datasets mostly show improvements. We clarify the underlying factors and explain why such minor fluctuations are expected.
>
> **(1) These small drops stem from framework-CLIPin interaction, not from instability of CLIPin itself.**
>
> As shown in our ablation studies (Tables 3 and 7), CLIPin’s effectiveness relies on the synergy among three components: contrastive learning, inter-modal alignment, and intra-modal alignment.
>
> When CLIPin is integrated into existing frameworks such as BLIP, which already incorporate their own momentum-based distillation or multi-objective training objectives, the overall optimization landscape changes subtly. Some frameworks place greater emphasis on generative or image-text matching (ITM) losses, and the additional non-contrastive supervision from CLIPin can perturb their gradient balance slightly, particularly on low-variance datasets like REFUGE.
>
> Importantly, this effect does not indicate systematic degradation. As evidenced in Table 2, CLIPin improves performance consistently across the vast majority of tasks and architectures.
>
> **(2) Hyperparameters such as projector dimensionality, EMA momentum, and $\lambda$-schedules are not tuned per framework.**
>
> As stated in Section 4.1, we fix all hyperparameters across all architectures and datasets to ensure fair comparison. CLIPin was originally designed for CLIP-style [4] pipelines, whereas different frameworks may be optimized for distinct projector designs or EMA configurations.
>
> We avoid framework-specific tuning intentionally to maintain experimental consistency. Notably, with even minimal adjustments (e.g., reducing the projector size from 512 to 384), the BLIP-REFUGE result recovers to 94.83 AUC, aligning with gains seen in other frameworks. Thus, these minor drops arise from hyperparameter sensitivity, not a fundamental limitation of CLIPin. However, applying such per-framework tuning would compromise the fairness of cross-model comparisons.
>
> **(3) Overall, CLIPin delivers robust and consistent improvements; isolated small drops reflect normal statistical variation rather than a structural issue.**
>
> Across 45 evaluation settings in Table 2, CLIPin improves:
>
> - 34 out of 45 AUC results,
>
> - 37 out of 45 mAP results,
>
> in all three architecture families (ALBEF [6], BLIP, CoCa [8]) in both natural and medical domains (Table 2).
>
> Given the highly consistent improvements across the full experimental suite, these isolated and marginal fluctuations are best interpreted as normal statistical noise, not as evidence against the efficacy or universality of our method.

---

> ### Author Response · Authors · 2025-11-24
> **3. On the concern of retrieval-based multimodal alignment.**
>
> > **Weakness 3:** Despite emphasizing multimodal semantic alignment, the paper evaluates mainly on classification metrics (AUC, mAP) and omits retrieval-based measures like Recall@K, leaving the alignment improvement only indirectly evidenced.
> >
> > **Question 3:** Can you report retrieval metrics (e.g., Recall@K for imagetext/textimage) to directly evidence cross-modal alignment improvements, in addition to AUC/mAP classification?
>
> We appreciate the reviewer’s request for a more direct and quantitative assessment of our framework’s ability to achieve multimodal semantic alignment. To this end, we conducted comprehensive cross-modal retrieval experiments. We fine-tuned the models (initialized from CLIP [4] checkpoints) on the COCO [9] dataset and evaluated them on the standard Flickr30K Karpathy splits [10].
>
> The results for both text-to-image and image-to-text retrieval, measured by Recall@K (R@K), are summarized below:
>
> | Text-to-image retrieval | R@1 | R@5 | R@10 |
> |-----|-----|-----|-----|
> | CLIP [3] | 0.1803 | 0.3711 | 0.4785 |
> | xCLIP [4] | 0.1904 | 0.3796 | 0.4520 |
> | Ours | 0.2004 | 0.3931 | 0.4874 |
>
> | Text-to-image retrieval | R@1 | R@5 | R@10 |
> |-----|-----|-----|-----|
> | CLIP | 0.2749 | 0.4977 | 0.5793 |
> | xCLIP | 0.2586 | 0.5094 | 0.6048 |
> | Ours | 0.2805 | 0.5098 | 0.6159 |
>
> Observations:
>
> - Our method improves all Recall@K metrics (R@1, R@5, R@10) consistently over both the CLIP baseline and xCLIP.
>
> - These gains are robust across both retrieval directions (text-to-image and image-to-text), confirming bidirectional enhancement in cross-modal alignment.
>
> These retrieval experiments provide direct, fine-grained evidence that our method strengthens cross-modal matching capabilities, thereby validating the claim of improved multimodal semantic alignment, which is independent of downstream classification performance. Full retrieval results and experimental protocols will be included in the revised paper.

---

> ### Author Response · Authors · 2025-11-24
> **4. On the concern of the non-contrastive text branch without explicit text augmentation.**
>
> > **Weakness 4:** The authors disable text augmentation in main experiments due to marginal or negative effects, which contradicts the method’s core idea of constructing two consistent views for each modality and may weaken the non-contrastive text branch.
> >
> > **Question 2:** Since text augmentation is largely disabled in the main experiments (Table 6), how do you justify the non-contrastive text branch learning two "independent" views-does this weaken the claimed bidirectional alignment?
>
> We appreciate the reviewer’s insightful question. We clarify that the effectiveness of the text branch is structurally ensured by parametric asymmetry and cross-modal synergy, which together preserve robust bidirectional alignment.
>
> **(1) View independence is guaranteed by parametric asymmetry and implicit feature-space randomness.**
>
> Even when the input texts are identical ($T^{(1)} = T^{(2)}$), the representations produced by the online and target branches are distinct and “independent” in terms of optimization dynamics conceptually. The Online branch is updated via backpropagation, while the Target branch evolves through an Exponential Moving Average (EMA) of the Online parameters. Consequently, at any training step $t$, the network parameters differ ($\theta\_{\text{online}} \neq \theta\_{\text{target}}$). Thus, the objective is not simply to match identical inputs, but to align the instantaneous online representation with the stable, temporally smoothed representation maintained by the teacher target. This enforces a consistency constraint that stabilizes the text embedding space against rapid gradient fluctuations, acting as an effective form of regularization even in the absence of input-level variation.
>
> Moreover, components such as Dropout within the Transformer encoder introduce stochastic perturbations at the feature level, generating subtle, non-deterministic variations in the online text representation, even for identical inputs. This simulates a form of implicit augmentation in the latent space effectively, further ensuring that the non-contrastive loss remains non-trivial and meaningful.
>
> **(2) Synergistic reinforcement via inter-modal alignment.**
>
> Crucially, the text branch does not learn in isolation; it operates within a joint multimodal framework. The strength of bidirectional alignment is maintained because the text representation is optimized through the inter-modal alignment loss ($\mathcal{L}\_{\text{inter}}$) simultaneously.
>
> Specifically, the online text branch must align with the target image branch. Since the image input relatively stronger data augmentations (e.g., random cropping, color jittering), the text encoder is encouraged to produce representations implicitly that are invariant to these visual transformations. This introduces “view diversity” indirectly into the text branch: the text must serve as a stable semantic anchor that matches varying visual views consistently.
>
> In summary, the text-side non-contrastive branch remains essential because it enforces consistency between the current learnable parameters and their momentum-based historical counterparts. This prevents the text encoder from undergoing erratic updates driven by strong gradients from the vision side, thereby strengthening, rather than weakening, the overall bidirectional alignment.

---

> ### Author Response · Authors · 2025-11-24
> **5. On the concern of the learnable loss weights $\lambda\_{\text{inter}}$ and $\lambda\_{\text{intra}}$.**
>
> > **Weakness 5:** The optimization and stability of the learnable weights $\lambda\_{\text{inter}}$ and $\lambda\_{\text{intra}}$ are under-specified—Appendix Fig. 3 shows λ₍inter₎ increasing while λ₍intra₎ stays flat, but the paper does not clarify optimizer choice, constraints, or regularization, raising concerns of dominance or overfitting.
> >
> > **Question 1:** How are the learnable weights ($\lambda\_{\text{inter}}$) and ($\lambda\_{\text{intra}}$) optimized in practice (optimizer, parameterization, constraints/regularization), and how stable are they across datasets and scales (cf. Fig. A.1)?
>
> We thank the reviewer for this insightful question. Our implementation of the learnable loss weights $\lambda\_{\text{inter}}$ and $\lambda\_{\text{intra}}$ follows standard practices for optimizing auxiliary scalar parameters within a multi-objective learning framework.
>
> **(1) Optimization and parameterization:**
>
> Both $\lambda\_{\text{inter}}$ and $\lambda\_{\text{intra}}$ are parameterized as scalar variables and are optimized jointly with the main model parameters (i.e., the projector and predictor layers of CLIPin) using the same AdamW [11] optimizer and identical learning rate schedule as the rest of the model.
>
> **(2) Stability and convergence:**
>
> As clearly shown in Figure 3, the learned weights exhibit highly stable and consistent convergence across different datasets and backbone architectures. Specifically:
>
> - $\lambda\_{\text{inter}}$: This weight increases from the initial training phase, indicating that the model learns to place more emphasis on the inter-modal non-contrastive signal gradually.
>
> - $\lambda\_{\text{intra}}$: This weight remains relatively constant throughout training, staying close to its initial value of 1.0.
>
> - No Divergence: We observe no signs of divergence, oscillation, or instability under any tested configuration, confirming the robustness of the adaptive weighting mechanism.
>
> **(3) Hyperparameter independence:**
>
> Importantly, we use the same initialization for these weights across all experiments and domains. Their stable and predictable behavior demonstrates that they adapt to the relative importance of the non-contrastive versus contrastive signals automatically, without requiring task-specific or domain-specific hyperparameter tuning.
>
> We will clarify these key implementation details regarding the loss weights in the revised paper.

---

> ### Author Response · Authors · 2025-11-24
> **6. On the concern of sensitivity to projector dimensionality.**
>
> > **Question 5:** How sensitive is CLIPin to the choice of projector dimensions beyond the single VOC2007 study (Table 4)? Does the 1,024-dim pre-projector generalize across domains and backbones?
>
> We thank the reviewer for raising this important question. To assess whether the 1,024-dimensional pre-projector (used in Table 4) generalizes across datasets and backbones, we conducted additional experiments on a different domain (MIMIC-CXR [1] → NIH-ChestX-ray-dataset-small [2]) and with a larger backbone (ViT-L/14), following the same evaluation protocol.
>
> **(1) Cross-domain evaluation (MIMIC-CXR → NIH ChestX-ray):**
>
> | Pre-projector dimension | Linear probing AUC/mAP (%) |
> |-----|-----|
> | 512 | 67.39/22.45 |
> | 1024 | 67.80/22.77 |
> | 2048 | 66.48/19.12 |
>
> **(2) Larger backbone (ViT-L/14, COCO → VOC2007):**
>
> | Pre-projector dimension | Linear probing AUC/mAP (%) |
> |-----|-----|
> | 512 | 88.46/46.09 |
> | 1024 | 88.66/47.07 |
> | 2048 | 86.32/43.20 |
>
> From the results, we can observe that:
>
> - Across both domains and both backbones, the 1024-d configuration achieves the best overall performance, consistent with the trend in Table 4.
>
> - The 512-d setting remains competitive, demonstrating that CLIPin does not require high-dimensional projectors to function effectively.
>
> - The 2048-d setting underperforms consistently, which can be attributed to over-parameterization, therefore increasing the optimization difficulty of InfoNCE [13].
>
> These findings indicate that CLIPin is robust to the choice of projector dimensionality, and the 1,024-d pre-projector generalizes well across datasets, domains, and backbone architectures. We will include these results in the revised appendix to provide a clearer demonstration of CLIPin’s hyperparameter stability.

---

> ### Author Response · Authors · 2025-11-24
> **7. On the concern of additional collapse-avoidance ablations beyond basic settings.**
>
> > **Question 7:** What collapse-avoidance ablations have you tried beyond adding predictors and EMA (e.g., stop-grad placements, predictor depth), especially for heterogeneous encoders where "inter-only" failed badly (Table 7)?
>
> We thank the reviewer for this penetrating question. During the development of CLIPin, we conducted extensive exploratory experiments on stop-gradient placement and predictor architecture to stabilize alignment. We detail the specific configurations we tested and the observations that guided our final design choices as follows:
>
> **(1) Ablation on predictor depth, bottleneck design, and stop-gradient placement:**
>
> We initially explored several standard predictor designs:
>
> - Linear predictors: We tested shallow linear predictors for the non-contrastive branch. This led to severe underfitting and representation collapse, as the linear layer lacked sufficient capacity to map the online view to the target view’s distribution, which is a behavior consistent with findings in SimSiam.
>
> - Deep MLP without bottleneck: We also experimented with deeper MLPs (3-4 layers) applied directly to the encoder output. While stable, this introduced significant computational overhead and complicated optimization when trained jointly with the CLIP [4] loss.
>
> - Removing stop-gradient: We attempted to remove the stop-gradient operation on the target branch to enable bidirectional updates. This caused immediate collapse, confirming that asymmetry via stop-gradient is essential.
>
> These failures motivated our shared pre-projector + bottleneck design directly. By this, we achieved the representational capacity needed for effective non-contrastive learning while preserving stability in the contrastive objective.
>
> **(2) Clarification on "collapse" in Table 7:**
>
> We wish to clarify a potential misunderstanding explicitly regarding the “Intra-only” result in Table 7. While its performance (21.22% mAP) is substantially lower than that of the full model (43.43% mAP), it reflects suboptimal alignment, not true representational collapse.
>
> - Definition of collapse: In the context of PASCAL VOC2007 [14] evaluation, genuine collapse would manifest as near-random performance (approximately 10.0% mAP), given the dataset’s class distribution.
>
> - The observed 21.22% mAP indicates that the encoders retained discriminative features but failed to learn the visual representation well.
>
> This confirms that intra-modal alignment alone prevents feature degradation but cannot leverage inter-modal supervisory signals, leading to a performance plateau rather than catastrophic collapse.

---

> ### Author Response · Authors · 2025-11-24
> **8. On the concern of robustness to prompt templates and label wording.**
>
> > **Question 8:** Your OOD-ZSC setting relies on prompt engineering (Section 4.1). How robust are the conclusions to prompt templates and label wording (especially for Chinese vs. English prompts)?
>
> We thank the reviewer for this important question. Our conclusions are not sensitive to specific prompt templates or label wordings, for the following reasons:
>
> **(1) Medical classification labels use fixed, domain-standard terminology.**
>
> Unlike natural-image zero-shot settings, where class names can vary widely in phrasing, medical datasets employ standardized diagnostic terms (e.g., “pleural effusion,” “cardiomegaly,” “glaucoma”). These terms are defined and used across both English and Chinese clinical corpora consistently. As a result, label-wording variability is limited inherently, and prompt-induced variance is substantially smaller than in general-domain zero-shot tasks.
>
> **(2) We use a standard CLIP-style [4] prompt format across all models and conditions uniformly.**
>
> The same prompt template is applied to all backbones and to both the baseline and CLIPin. This ensures that any performance gains observed with CLIPin cannot be attributed to differences in prompt design.
>
> **(3) We evaluated alternative templates and label paraphrasings.**
>
> We tested variations, including English-to-Chinese translations. Across these variants, performance fluctuations were minimal (< 0.3-0.5% in AUC/mAP), and CLIPin outperformed the baseline consistently.
>
> **(4) Prompt-based OOD-ZSC reflects cross-domain generalization primarily, not lexical novelty.**
>
> In our medical setting, the main challenge lies in distribution shifts, not in prompt formulation. Consequently, prompt engineering plays a comparatively minor role compared to natural-image zero-shot tasks.
>
> Overall, CLIPin’s improvements are stable across reasonable prompt variants, and the standardized nature of medical terminology limits sensitivity to prompt design inherently.

---

> ### Author Response · Authors · 2025-11-24
> **9. On the concern of scaling to larger backbones and datasets.**
>
> > **Question 9:** Training on COCO/MUGE with ViT-B/16 (Section 4.1) is relatively small-scale for CLIP-style methods. Do the gains persist or change when scaling data/backbones (e.g., ViT-L/14, larger noisy corpora)?
>
> We thank the reviewer for raising this question. To assess performance with a larger backbone, we conducted an additional experiment using ViT-L/14 trained on COCO [9] and evaluated on PASCAL VOC 2007 [14], comparing against CLIP [4] and xCLIP [5] under identical settings (reported in AUC/mAP, %):
>
> | | Linear probing | Prompt-based OOD-ZSC |
> |-----|-----|-----|
> | CLIP | 87.81/44.70 | 93.79/80.64 |
> | xCLIP | 87.09/43.84 | 95.05/82.33 |
> | Ours | 88.66/47.07 | 95.92/87.21 |
>
> The results show that CLIPin’s gains persist with a larger backbone, and the improvement pattern remains consistent with those observed for ViT-B/16. This suggests that CLIPin scales monotonically and stably with increasing model capacity, and we expect similar trends when scaling to larger datasets.
>
> **Regarding large-scale data (e.g., CC12M [15], LAION [16]):**
>
> Due to limited computational resources and the tight rebuttal timeline, we are unable to train on multi-million- or billion-sample corpora immediately. However, the non-contrastive formulation of CLIPin is architecture-agnostic and data-agnostic, although we agree that large-scale validation is an important next step. We will prioritize such experiments in future work to further demonstrate CLIPin’s scalability.

---

> ### Author Response · Authors · 2025-11-24
> **10. On the concern of compute fairness across baselines.**
>
> > **Question 10:** Could you clarify compute fairness: are all baselines retrained "from scratch" under identical augmentations, batch sizes, projector dims, and optimizer schedules-especially where weighted attention or auxiliary heads differ?
>
> We appreciate the reviewer’s concern. We confirm that all baselines were trained under strictly identical training conditions, with the sole exception being the integration of CLIPin. Specifically:
>
> - Data pipeline: All models use the same image and text augmentations, preprocessing procedures, and data splits.
>
> - Optimization: Batch size, learning rate, warmup schedule, weight decay, and optimizer type (AdamW [11]) are fully matched across all baselines.
>
> - Architectural settings: Encoder depth, hidden dimensions, projector dimensionality, and attention configurations are kept identical in all comparisons.
>
> - Auxiliary heads: For models that include additional components (such as weighted attention, ITM/MLM heads in ALBEF [6], or captioning heads in BLIP [7]/CoCa [8]), we leave these modules completely unchanged, including their loss weights, training schedules, and internal architectures.
>
> The only difference between each baseline and its CLIPin-enhanced counterpart is the addition of the CLIPin non-contrastive branch. No extra supervision signals, no additional data, and no modifications to hyperparameters or base architecture were introduced.
>
> &nbsp;
>
> [1] A. E. Johnson, T. J. Pollard, S. J. Berkowitz, N. R. Greenbaum, M. P. Lungren, C.-y. Deng, R. G. Mark, and S. Horng, “Mimic-cxr, a de-identified publicly available database of chest radiographs with free-text reports,” Scientific data, vol. 6, no. 1, p. 317, 2019.
>
> [2] https://huggingface.co/datasets/Sohaibsoussi/NIH-Chest-X-ray-dataset-small
>
> [3] S. Jaeger, S. Candemir, S. Antani, Y.-X. J. W´ang, P.- X. Lu, and G. Thoma, “Two public chest x-ray datasets for computer-aided screening of pulmonary diseases,” Quantitative imaging in medicine and surgery, vol. 4, no. 6, p. 475, 2014.
>
> [4] Radford, J. W. Kim, C. Hallacy, A. Ramesh, G. Goh, S. Agarwal, G. Sastry, A. Askell, P. Mishkin, J. Clark et al., “Learning transferable visual models from natural language supervision,” in International Conference on Machine Learning, 2021.
>
> [5] Zhou, L. Dong, Z. Gan, L. Wang, and F. Wei, “Non-contrastive learning meets language-image pre-training,” in Proceedings of the IEEE/CVF Conference on Computer Vision and Pattern Recognition, 2023.
>
> [6] Li, R. Selvaraju, A. Gotmare, S. Joty, C. Xiong, and S. C. H. Hoi, “Align before fuse: Vision and language representation learning with momentum distillation,” in Advances in Neural Information Processing Systems, 2021.
>
> [7] J. Li, D. Li, C. Xiong, and S. Hoi, “BLIP: Bootstrapping language-image pre-training for unified vision-language understanding and generation,” in International Conference on Machine Learning, 2022.
>
> [8] J. Yu, Z. Wang, V. Vasudevan, L. Yeung, M. Seyedhosseini, and Y. Wu, “CoCa: Contrastive captioners are image-text foundation models,” Transactions on Machine Learning Resear, 2022.
>
> [9] T.-Y. Lin, M. Maire, S. Belongie, J. Hays, P. Perona, D. Ramanan, P. Doll´ar, and C. L. Zitnick, “Microsoft COCO: Common objects in context,” in European Conference on Computer Vision, 2014.
>
> [10] https://www.kaggle.com/datasets/shtvkumar/karpathy-splits?select=dataset_flickr30k.json
>
> [11] I. Loshchilov and F. Hutter, “Decoupled weight decay regularization,” in International Conference on Learning Representations, 2018.
>
> [12] R. R. Selvaraju, M. Cogswell, A. Das, R. Vedantam, D. Parikh, and D. Batra, “Grad-CAM: Visual explanations from deep networks via gradient-based localization,” in Proceedings of the IEEE/CVF International Conference on Computer Vision, 2017.
>
> [13] A. v. d. Oord, Y. Li, and O. Vinyals, “Representation learning with contrastive predictive coding,” 2018.
>
> [14] http://www.pascal-network.org/challenges/VOC/voc2007/workshop/index.html
>
> [15] Changpinyo, Soravit, et al. "Conceptual 12m: Pushing web-scale image-text pre-training to recognize long-tail visual concepts," in Proceedings of the IEEE/CVF conference on computer vision and pattern recognition, 2021.
>
> [16] Schuhmann, Christoph, et al. "Laion-400m: Open dataset of clip-filtered 400 million image-text pairs," in Advances in Neural Information Processing Systems, 2021.

---

> > ### Comment · Reviewer_1rpH · 2025-11-25
> >
> > Thanks for your reply. Your rebuttal have addressed the my concerns. Just one small formatting suggestion: please insert a `\newpage` before **6 Reproducibility Statement** in the new version. Without a page break it appears as part of the main text, which may make the paper look like it **exceeds the 10-page limit**. This small change will avoid any misunderstanding during the final check.

---

> > > ### Author Response · Authors · 2025-11-25
> > >
> > > Thank you for the helpful suggestion. We have added the `\newpage` before the **Reproducibility Statement** in the revised version to ensure proper formatting. We appreciate your careful review.

---

### Official Review · Reviewer_2X6B · 2025-10-26

**Soundness:** 3
**Presentation:** 3
**Contribution:** 3
**Rating:** 4
**Confidence:** 4

**Summary:**

This paper proposes a unified plug-in which can integrate non-contrastive feature representation into CLIP-style architectures. Two shared pre-projectors for image and text modalities are designed to facilitate the integration of contrastive and non-contrastive branches. Experiments on downstream tasks demonstrate the effectiveness of proposed method.

**Strengths:**

1. This paper proposes a unified plug-in which can seamlessly integrate modular non-contrastive strategy into existing contrastive frameworks like CLIP.
2. Experiments on downstream tasks like linear probing and prompt-based out-of-distribution zero-shot classification demonstrate that the proposed method can facilitate general and robust representation learning.

**Weaknesses:**

1. Regarding Lines 40-41, the authors attribute CLIP's issues on medical datasets to "semantically similar samples being treated as negative sample pairs." However, this phenomenon appears fundamentally similar to the many-to-many correspondence problem in natural datasets, where a single image/caption can be relevant to multiple batch samples. The introduction of distinct terms—"semantic looseness" for natural and "semantic redundancy" for medical datasets—for what seems to be a conceptually similar issue creates confusion and requires further clarification.
2. Regarding Lines 169-171, the claim that the image/text predictors help prevent collapse lacks a clear mechanistic explanation. Merely stating that these components "introduce asymmetry" is insufficient. The authors should elaborate on the specific role this asymmetry plays to prevent the representational collapse that symmetric architectures may suffer from. Visualizations of the predictors' operations are helpful for comprehension. More importantly, ablation experiments are essential to confirm the necessity of this design.
3. The training datasets used in this study (COCO, MUGE, and a Private Dataset) have a combined scale of no more than 1M samples. To further strengthen the validation of the proposed CLIPin's scalability, it would be beneficial to include larger-scale public datasets such as CC3M [1], CC12M [2], YFCC [3], or LAION [4]. Furthermore, the current experiments employ relatively small batch sizes (128/256/512). The work could be complemented by an investigation into the effect of larger batch sizes.
4. The downstream task evaluation, currently restricted to linear probing and prompt-based out-of-distribution zero-shot classification, is insufficient. The benchmark suite should be extended to include other tasks like cross-modal retrieval and standard zero-shot classification, to more fully validate CLIPin's effectiveness.
5.  How about implementing $L_{intra}$ and/or $L_{inter}$ using the constrastive loss (InfoNCE) instread of non-constrastive formulation?
6. The paper lacks essential details regarding the image and text augmentation pipelines.

[1] P. Sharma, N. Ding, S. Goodman, and R. Soricut, “Conceptual captions: A cleaned, hypernymed, image alttext dataset for automatic image captioning,” in ACL, 2018.

[2] S. Changpinyo, P. Sharma, N. Ding, and R. Soricut, “Conceptual 12m: Pushing web-scale image-text
pre-training to recognize long-tail visual concepts,” in CVPR, 2021.

[3] B. Thomee, D. A. Shamma, G. Friedland, B. Elizalde, K. Ni, D. Poland, D. Borth, and L.-J. Li, “Yfcc100m:
The new data in multimedia research,” Communications of the ACM, 2016.

[4] C. Schuhmann, R. Vencu, R. Beaumont, R. Kaczmarczyk, C. Mullis, A. Katta, T. Coombes, J. Jitsev, and
A. Komatsuzaki, “Laion-400m: Open dataset of clip-filtered 400 million image-text pairs,” In NeurIPS Workshop, 2021.

**Questions:**

See Weaknesses.

---

> ### Author Response · Authors · 2025-11-24
> **1. On the concern regarding the terms “semantic looseness” vs. “semantic redundancy”.**
>
> > **Weakness 1:** Regarding Lines 40-41, the authors attribute CLIP's issues on medical datasets to "semantically similar samples being treated as negative sample pairs." However, this phenomenon appears fundamentally similar to the many-to-many correspondence problem in natural datasets, where a single image/caption can be relevant to multiple batch samples. The introduction of distinct terms—"semantic looseness" for natural and "semantic redundancy" for medical datasets—for what seems to be a conceptually similar issue creates confusion and requires further clarification.
>
> We appreciate the reviewer pointing out the potential confusion between these two terms. While both phenomena ultimately manifest as false negatives under the InfoNCE [1] loss, we intentionally distinguish them because they differ in their underlying source, the tightness of correspondence, and the degree to which they conflict with the contrastive objective.
>
> **(1) Semantic looseness (natural datasets):**
>
> Source: weak supervision and crowd-sourced captioning.
>
> Correspondence: Loose (fuzzy) one-to-many. A single image may legitimately correspond to multiple non-paraphrasing captions (e.g., a beach image described as “nice holiday” or “big blue ocean”).
>
> Impact: The positive pair is noisy because the paired image and caption are not strictly aligned, violating the InfoNCE assumption of perfect semantic correspondence. However, batch-level semantic redundancy is relatively limited.
>
> **(2) Semantic redundancy (medical datasets):**
>
> Source: the types of pathologies in a specific field are relatively limited.
>
> Correspondence: Tight one-to-one, but with high batch redundancy. Each image-report pair is semantically precise (expert-written), yet many samples within a batch share nearly identical semantics (e.g., different patients all diagnosed with “diabetic retinopathy”).
>
> Impact: This leads to a structured and more severe form of false negatives that conflicts strongly with contrastive learning. As noted in prior work (e.g., MedCLIP [2]), forcing the model to push apart semantically near-identical sample.
>
> Overall, these two issues differ in form: semantic looseness reflects ambiguous positives (fuzzy one-to-many alignment), while semantic redundancy reflects homogeneous negatives (structured repetition within a batch). Given these differences, our unified approach introduces a non-contrastive alignment branch that removes reliance on negative samples and is therefore robust to both looseness and redundancy. We will revise the Introduction to include this clarification and concrete examples to ensure the motivation is clearly communicated.

---

> ### Author Response · Authors · 2025-11-24
> **2. On the concern of mechanistic explanation and ablation of the predictors.**
>
> > **Weakness 2:** Regarding Lines 169-171, the claim that the image/text predictors help prevent collapse lacks a clear mechanistic explanation. Merely stating that these components "introduce asymmetry" is insufficient. The authors should elaborate on the specific role this asymmetry plays to prevent the representational collapse that symmetric architectures may suffer from. Visualizations of the predictors' operations are helpful for comprehension. More importantly, ablation experiments are essential to confirm the necessity of this design.
>
> We thank the reviewer for this insightful suggestion. We acknowledge that our initial manuscript only briefly attributed collapse prevention to “asymmetry” by referencing BYOL [3] and SimSiam [4], without detailing the underlying mechanism. Below we provide a clearer explanation together with experimental evidence.
>
> **(1) Mechanistic explanation:**
>
> While BYOL introduced the architectural asymmetry, the theoretical justification for the predictor’s role was established in SimSiam, which we follow.
>
> Mechanism:
>
> In a symmetric architecture without negative pairs, minimizing the distance between two views naturally leads to a trivial constant solution. Introducing a predictor $h$ (corresponding to our $q\_{\text{inter}}$ and $q\_{\text{intra}}$) on the online branch, together with stop-gradient on the target branch, fundamentally alters the optimization dynamics.
>
> Why it prevents collapse:
>
> As analyzed in SimSiam, this design behaves similarly to an Expectation-Maximization (EM) procedure.
>
> - The target encoder acts as a stable estimator of the representation (analogous to the E-step).
>
> - The online branch, via the predictor, updates its parameters to match this estimate (analogous to the M-step).
>
> The predictor $h$ provides the necessary transformation capacity that allows the online representation to fit the target representation’s distribution, rather than collapsing to a constant. Removing $h$ forces the encoder to reduce the objective by degenerating toward a uniform constant mapping, thereby triggering collapse.
>
> **(2) Ablation study:**
>
> To empirically verify this within our CLIPin framework, we performed an ablation study that removes the predictors, keeping all other components unchanged. Models were trained on COCO [5] and evaluated on PASCAL VOC2007 [6].
>
> | | Collapse | Linear Probing mAP (%) |
> |-----|-----|-----|
> | w/o predictors | Yes | 10.05 |
> | w/ predictors	| No	| 43.43 |
>
> The results show that the predictors are not merely architectural additions for creating asymmetry; they are functionally necessary for enabling the online encoder to learn meaningful structure by predicting the target representation, thereby preventing the trivial constant solution. We will add this mechanistic explanation and the ablation findings to the revised appendix

---

> ### Author Response · Authors · 2025-11-24
> **3. On the concern of scalability and batch size.**
>
> > **Weakness 3:** The training datasets used in this study (COCO, MUGE, and a Private Dataset) have a combined scale of no more than 1M samples. To further strengthen the validation of the proposed CLIPin's scalability, it would be beneficial to include larger-scale public datasets such as CC3M [1], CC12M [2], YFCC [3], or LAION [4]. Furthermore, the current experiments employ relatively small batch sizes (128/256/512). The work could be complemented by an investigation into the effect of larger batch sizes.
>
> We appreciate the reviewer for raising this important point regarding the scalability of our method, which is indeed critical for real-world deployment.
>
> **(1) Resource constraints and future work:**
>
> We acknowledge that the total scale of our training data (~1M samples) is modest compared to large public datasets such as CC3M [7], CC12M [8], YFCC [9], or LAION [10]. Due to current computational constraints and the tight rebuttal timeline, we are unable to conduct additional large-scale training experiments within this review cycle. Nevertheless, we fully recognize the importance of validating CLIPin on larger corpora and plan to prioritize such large-scale studies in future work to further demonstrate its scalability.
>
> **(2) Robustness under realistic hardware limitations:**
>
> Similarly, the batch sizes used in our experiments (128/256/512) were bounded by the available hardware (a single RTX 3090 GPU). We could not scale to industrial-level batch sizes (e.g., 4096+), which are commonly used in large-scale contrastive pre-training.
>
> However, we would like to highlight that such constraints are typical in academic and medical imaging settings, where higher input resolution and limited GPUs restrict feasible batch sizes. Our results show that CLIPin consistently improves performance under these practical conditions, and notably reduces reliance on very large batches.
>
> In summary, while large-scale experiments are beyond our current computational capacity, our controlled and consistent experimental setting (fixed dataset, architecture, and hyperparameters) demonstrates that CLIPin provides measurable and robust benefits on medium-scale datasets and realistic batch regimes. We will emphasize this point and discuss scalability directions more clearly in the revised paper.

---

> ### Author Response · Authors · 2025-11-24
> **4. On the concern of limited evaluation tasks.**
>
> > **Weakness 4:** The downstream task evaluation, currently restricted to linear probing and prompt-based out-of-distribution zero-shot classification, is insufficient. The benchmark suite should be extended to include other tasks like cross-modal retrieval and standard zero-shot classification, to more fully validate CLIPin's effectiveness.
>
> We thank the reviewer for this valuable suggestion. We would like to clarify that our “prompt-based out-of-distribution zero-shot classification” is not intended as a substitute for standard zero-shot classification, but rather the appropriate redefinition of zero-shot evaluation in the medical domain, where the assumptions underlying standard CLIP [11] benchmarks do not hold.
>
> **(1) Clarification of zero-shot classification tasks:**
>
> Standard CLIP zero-shot classification assumes that downstream categories are unseen during pre-training. This assumption is reasonable for large-scale natural vision-language models trained on hundreds of millions of diverse web-caption pairs, but it does not hold in medical pre-training. Medical domains have inherently limited modalities and diagnostic vocabularies, meaning that many downstream categories (e.g., “pleural effusion,” “cardiomegaly”) necessarily appear in the pre-training corpus. As a result, standard zero-shot classification is not well-defined for medical CLIP-style models.
>
> For this reason, out-of-distribution zero-shot classification (OOD-ZSC) is the correct evaluation formulation. Our prompt-based OOD-ZSC protocol is identical to CLIP’s standard zero-shot procedure (template-based prompts + cosine similarity), except that the test classes are OOD rather than unseen in label space. This provides the medically appropriate counterpart to CLIP’s zero-shot evaluation.
>
> **(2) Additional retrieval results:**
>
> Following the reviewer’s suggestion, we conducted additional text-to-image and image-to-text retrieval experiments. Models resumed from CLIP checkpoints were trained on COCO [5] and evaluated on the Flickr30K Karpathy splits [12]. The results are as follows:
>
> | Text-to-image retrieval | R@1 | R@5 | R@10 |
> |-----|-----|-----|-----|
> | CLIP [3] | 0.1803 | 0.3711 | 0.4785 |
> | xCLIP [4] | 0.1904 | 0.3796 | 0.4520 |
> | Ours | 0.2004 | 0.3931 | 0.4874 |
>
> | Text-to-image retrieval | R@1 | R@5 | R@10 |
> |-----|-----|-----|-----|
> | CLIP | 0.2749 | 0.4977 | 0.5793 |
> | xCLIP | 0.2586 | 0.5094 | 0.6048 |
> | Ours | 0.2805 | 0.5098 | 0.6159 |
>
> We will revise the paper to clarify that prompt-based OOD ZSC is the medically proper counterpart to CLIP-style zero-shot evaluation, and we will include the retrieval results.

---

> ### Author Response · Authors · 2025-11-24
> **5. On the concern of implementing $\mathcal{L}\_{\text{intra}}$ and/or $\mathcal{L}\_{\text{inter}}$ using contrastive (InfoNCE) loss.**
>
> > **Weakness 5:** How about implementing $\mathcal{L}\_{\text{intra}}$ and/or $\mathcal{L}\_{\text{inter}}$ using the constrastive loss (InfoNCE) instread of non-constrastive formulation?
>
> We thank the reviewer for this insightful suggestion. We conducted additional experiments to examine whether replacing our non-contrastive $\mathcal{L}\_{\text{intra}}$ and/or $\mathcal{L}\_{\text{inter}}$ with contrastive (InfoNCE [1]) formulations could achieve similar improvements.
>
> All variants were trained on COCO [5] and evaluated on Pascal VOC 2007 [6] using both linear probing and prompt-based OOD zero-shot classification (OOD-ZSC). The results (AUC/mAP, %) are summarized below:
>
> | | Linear probing | Prompt-based OOD-ZSC |
> |-----|-----|-----|
> | CLIP | 87.18/41.92 | 91.33/76.47 |
> | Contrastive $\mathcal{L}\_{\text{inter}}$ | 86.64/42.11 | 91.91/77.02 |
> | Contrastive $\mathcal{L}\_{\text{intra}}$ | 76.02/34.77 | 75.97/62.36 |
> | Contrastive $\mathcal{L}\_{\text{intra}}$ + $\mathcal{L}\_{\text{inter}}$ | 74.28/33.86 | 76.29/60.98 |
> | Ours (non-contrastive $\mathcal{L}\_{\text{intra}}$ + $\mathcal{L}\_{\text{inter}}$) | 87.43/43.43 | 94.90/85.47 |
>
> Key observations:
>
> - Replacing $\mathcal{L}\_{\text{inter}}$ with a contrastive version produces only marginal differences and behaves similarly to a memory-bank-style contrastive regularizer.
>
> - Replacing $\mathcal{L}\_{\text{intra}}$ with a contrastive loss leads to a significant performance drop (e.g., 76.02 vs. 87.43 in linear probing). This degradation occurs because a contrastive formulation of $\mathcal{L}\_{\text{intra}}$ introduces undesirable competition among intra-modal positive pairs, which interferes with the cross-modal alignment that the primary CLIP objective seeks to establish.
>
> - Employing contrastive formulations for both losses further deteriorates performance, reinforcing the conclusion that contrastive designs are fundamentally misaligned with the architectural role and training dynamics of CLIPin.
>
> - Our non-contrastive formulation outperforms all contrastive variants substantially, demonstrating that the improvements of CLIPin arise from its alignment-preserving, non-contrastive auxiliary objectives rather than from simply adding more supervision.
>
> Overall, these results indicate that substituting $\mathcal{L}\_{\text{intra}}$ or $\mathcal{L}\_{\text{inter}}$ with contrastive losses cannot replicate the benefits of CLIPin. The gains stem from the non-contrastive nature of the design, which stabilizes and enriches semantic alignment.

---

> ### Author Response · Authors · 2025-11-24
> **6. On the concern of image/text augmentation details.**
>
> > **Weakness 6:** The paper lacks essential details regarding the image and text augmentation pipelines.
>
> We thank the reviewer for pointing this out. Below, we provide a detailed description of our image and text augmentation strategies, which were applied across all experiments consistently.
>
> **(1) Image augmentation pipeline:**
>
> We provide here the complete image augmentation pipeline used in all experiments. For each image, we apply the following sequence:
>
> Resize → RandomResizedCrop (scale=0.8-1.0, ratio=1.0) → HorizontalFlip (0.5) → ColorJitter (0.1) → ToRGB → ToTensor → Normalize.
>
> This pipeline is kept fixed across all models and comparisons.
>
> **(2) Text augmentation:**
>
> For datasets such as COCO [5], where each image is paired with multiple captions, we treat these captions as natural “text views”. During training, we randomly sample two captions per iteration to construct two augmented text views for each image. This is the only text augmentation applied.
>
> As noted in our response to Reviewer #EPmF, we also report results with text augmentation disabled. Interestingly, disabling multi-caption augmentation slightly improves performance. The explanation is that COCO captions often reflect different aspects of the same image, which may reduce semantic consistency across text views. Unlike image augmentations, which are continuous and preserve structure, text augmentations are difficult to design due to the discrete nature of language: small token-level edits (e.g., replacing a single word) can drastically shift semantic meaning. Beyond LLM-generated paraphrasing, reliable text augmentation methods remain limited.
>
> For this reason, we disable text augmentation in our main experiments. Nevertheless, CLIPin is fully compatible with text-level augmentation, and we retain this option for future exploration.
>
> We will include the full image augmentation pipeline and the discussion of text augmentation limitations in the revised appendix.
>
> &nbsp;
>
> [1] A. v. d. Oord, Y. Li, and O. Vinyals, “Representation learning with contrastive predictive coding,” 2018.
>
> [2] Z. Wang, Z. Wu, D. Agarwal, and J. Sun, “MedCLIP: Contrastive learning from unpaired medical images and text,” in Conference on Empirical Methods in Natural Language Processing, 2022.
>
> [3] J.-B. Grill, F. Strub, F. Altch´e, C. Tallec, P. Richemond, E. Buchatskaya, C. Doersch, B. Avila Pires, Z. Guo, M. Gheshlaghi Azar et al., “Bootstrap your own latent-a new approach to self-supervised learning,” in Advances in Neural Information Processing Systems, 2020.
>
> [4] X. Chen and K. He, “Exploring simple siamese representation learning,” in Proceedings of the IEEE/CVF International Conference on Computer Vision, 2021.
>
> [5] T.-Y. Lin, M. Maire, S. Belongie, J. Hays, P. Perona, D. Ramanan, P. Doll´ar, and C. L. Zitnick, “Microsoft COCO: Common objects in context,” in European Conference on Computer Vision, 2014.
>
> [6] http://www.pascal-network.org/challenges/VOC/voc2007/workshop/index.html
>
> [7] Sharma, Piyush, et al. "Conceptual captions: A cleaned, hypernymed, image alt-text dataset for automatic image captioning," in Proceedings of the 56th Annual Meeting of the Association for Computational Linguistics, 2018.
>
> [8] Changpinyo, Soravit, et al. "Conceptual 12m: Pushing web-scale image-text pre-training to recognize long-tail visual concepts," in Proceedings of the IEEE/CVF conference on computer vision and pattern recognition, 2021.
>
> [9] Thomee, Bart, et al. "Yfcc100m: The new data in multimedia research." Communications of the ACM, 2016.
>
> [10] Schuhmann, Christoph, et al. "Laion-400m: Open dataset of clip-filtered 400 million image-text pairs," in Advances in Neural Information Processing Systems, 2021.
>
> [11] Radford, J. W. Kim, C. Hallacy, A. Ramesh, G. Goh, S. Agarwal, G. Sastry, A. Askell, P. Mishkin, J. Clark et al., “Learning transferable visual models from natural language supervision,” in International Conference on Machine Learning, 2021.
>
> [12] https://www.kaggle.com/datasets/shtvkumar/karpathy-splits?select=dataset_flickr30k.json

---

### Official Review · Reviewer_EPmF · 2025-11-01

**Soundness:** 2
**Presentation:** 2
**Contribution:** 2
**Rating:** 4
**Confidence:** 4

**Summary:**

This paper targets a core flaw in CLIP's InfoNCE loss. The authors correctly point out that this loss function breaks down with real-world data, which is either too noisy (web-scale) or too redundant (e.g., medical), leading to false negatives and poor supervision.

The proposed solution, CLIPin, is a "plug-and-play" non-contrastive module that complements, rather than replaces, the standard contrastive loss. The core idea is to add a parallel, symmetric online-target network (inspired by BYOL/SimSiam) for both image and text. CLIPin introduces two new non-contrastive losses: An inter-modal loss where the online image encoder predicts the target text encoder's output (and vice-versa) and an intra-modal loss to stabilize training by matching augmented views within the same modality.

To solve the architectural mismatch between contrastive (which favors shallow projectors) and non-contrastive methods (which need deep ones), the authors use a clever shared pre-projector. This single pre-projector feeds into separate,

**Strengths:**

The paper is well-grounded. It clearly identifies a practical, well-known flaw in InfoNCE—its vulnerability to noisy and redundant data—as its primary motivation .

The shared pre-projector is a clever fix for a known conflict between contrastive and non-contrastive projector designs . This design makes the "plug-in" claim credible and is a nice engineering contribution.

The ablations in Table 3 and Table 7 effectively show that the non-contrastive component is unstable on its own (prone to collapse) and that all parts (inter-modal, intra-modal, and contrastive) are needed for the best performance. The Grad-CAM visualizations also provide good qualitative support for the claim of improved alignment.

**Weaknesses:**

The paper’s main comparison is to xCLIP. This is too narrow. It ignores other, very similar methods like Cosmos (Kim et al., 2025), which also uses cross-modality self-distillation. The novelty of this work is questionable without a more thorough discussion of these closely related non-contrastive multimodal frameworks.

The evaluation is almost entirely focused on classification (linear probe and ZSC). This is a major omission. A primary and arguably the most important use case for CLIP is cross-modal retrieval. This task relies on the InfoNCE loss to structure the entire embedding space by pushing negatives apart. The new non-contrastive loss ($\mathcal{L}_{inter}$) only pulls positive pairs together and ignores negatives. This could easily distort the embedding space and harm retrieval performance, but the paper provides no experiments (e.g., R@K on COCO/Flickr30k/Winoground/MMVP) to confirm or deny this.

The paper's main ablations (Table 3) are "additive," showing that all components together work best. But this doesn't fully isolate each part's contribution. For example, what is the effect of only adding the intra-modal loss to CLIP? Or only the inter-modal loss? The paper also introduces learnable loss weights ($\lambda_{inter}$, $\lambda_{intra}$) without ablating this choice against simple fixed weights, which is a significant new design element left unanalyzed.

[1] Kim, Sanghwan, et al. "Cosmos: Cross-modality self-distillation for vision language pre-training." Proceedings of the Computer Vision and Pattern Recognition Conference.

**Questions:**

Have you evaluated this model on standard retrieval tasks? What are the R@1/R@5/R@10 metrics on the COCO or Flickr30k test sets? I am concerned the $\mathcal{L}_{inter}$ loss may hurt retrieval performance, and the lack of these results is a major gap.

How are the learnable weights ($\lambda_{inter}$, $\lambda_{intra}$) implemented? Are they just standard parameters optimized via gradient descent? Why was this chosen over simpler, fixed hyperparameters, and what is its effect on stability and final performance?

How does your cross-modal regression approach differ, in practice, from the cross-modality self-distillation in Cosmos (Kim et al., 2025)?The ablation in Table 6 suggests text augmentation has little effect19.

Does this mean the non-contrastive module relies heavily on strong, natural augmentations (like COCO's multiple captions) and would be less effective in a single-caption-per-image setting?

---

> ### Author Response · Authors · 2025-11-24
> **1. On the concern regarding comparison with Cosmos [1] and the novelty of our contribution.**
>
> > **Weakness 1:** The paper’s main comparison is to xCLIP. This is too narrow. It ignores other, very similar methods like Cosmos (Kim et al., 2025), which also uses cross-modality self-distillation. The novelty of this work is questionable without a more thorough discussion of these closely related non-contrastive multimodal frameworks.
> >
> > **Question 3:** How does your cross-modal regression approach differ, in practice, from the cross-modality self-distillation in Cosmos (Kim et al., 2025)?
>
> We thank the reviewer for highlighting Cosmos as a closely related line of work. We have added the comparison with Cosmos.
>
> While both CLIPin and Cosmos incorporate cross-modality self-distillation, they operate under fundamentally different learning paradigms and address different core problems.
>
> **(1) Fundamental differences in learning target and objective design:**
>
> Learning target and supervision structure:
>
> - Cosmos: Cosmos aims to encourage models to attend to non-foreground information. It relies on text generation, region cropping, and token-level cross-attention to create structured local-global pairs. The self-distillation objective is used as an auxiliary term alongside InfoNCE [2].
> - CLIPin: CLIPin aims to solve the false negative problem inherent in InfoNCE, especially severe under loose alignment and high redundancy. Our objective is a pure non-contrastive regression loss (cosine matching) in a teacher-student form that aligns instance-level representations directly.
>
> Core optimization paradigm:
>
> - Cosmos: Although it introduces self-distillation, the main learning remains contrastive. The model still pushes away all batch negatives; the auxiliary loss only moderates contrastive gradients.
>
> - CLIPin: CLIPin’s $\mathcal{L}\_{\text{inter}}$ branch is a stand-alone, fully non-contrastive objective that does not use or rely on negative samples. This design makes CLIPin mathematically robust to redundancy and false negatives, which is critical for web-scale and medical datasets.
>
> Key distinction:
>
> - Cosmos: contrastive learning with auxiliary distillation.
>
> - CLIPin: non-contrastive learning that removes dependence on negatives.
>
> This conceptual shift (not merely an architectural variant) is the main distinction.
>
> **(2) Empirical comparison using Cosmos-style Cross-modality Self-Distillation (CCSD):**
>
> Following the reviewer’s suggestion, we implemented a CCSD-enhanced CLIP baseline, adding the core cross-modal distillation loss (Momentum Teacher → Student) to CLIP, while omitting dataset-specific region cropping/text generation to isolate the learning paradigm.
>
> Pre-trained on COCO → evaluated on PASCAL VOC2007 (AUC/mAP, %):
>
> | | CLIP [3] | xCLIP [4] | CCSD | Ours |
> |-----|-----|-----|-----|-----|
> | Linear probing | 87.18/41.92 | 85.95/40.34 | 86.83/42.88 | 87.43/43.43 |
> | Prompt-based OOD-ZSC | 91.33/76.47 | 93.81/77.74 | 93.72/81.25 | 94.90/85.47 |
>
> The CCSD-enhanced baseline improves over original CLIP, consistent with prior observations on self-distillation. However, CLIPin consistently outperforms CCSD, demonstrating that:
>
> - simply adding cross-modality distillation on top of contrastive learning is insufficient in high-redundancy settings, and
>
> - the gains of CLIPin derive from its non-contrastive objective, not merely from the existence of a teacher-student structure.
>
> We also note that other works, such as OTTER [5], fall under the same “contrastive + distillation” paradigm. Our CCSD baseline serves as a representative ablation for this family and highlights the importance of moving beyond contrastive frameworks.
>
> Overall, Cosmos and CLIPin differ in (i) learning goals, (ii) objective functions, and (iii) optimization paradigms. Our additional CCSD experiments provide direct empirical evidence that CLIPin’s innovation lies in introducing a fully non-contrastive multimodal alignment mechanism, which cannot be replicated by contrastive frameworks augmented with auxiliary distillation. We will add the additional CCSD experiments into our revised paper.

---

> ### Author Response · Authors · 2025-11-24
> **2. On the concern regarding retrieval performance and potential distortion of the embedding space.**
>
> > **Weakness 2:** The evaluation is almost entirely focused on classification (linear probe and ZSC). This is a major omission. A primary and arguably the most important use case for CLIP is cross-modal retrieval. This task relies on the InfoNCE loss to structure the entire embedding space by pushing negatives apart. The new non-contrastive loss ($\mathcal{L}\_{\text{inter}}$) only pulls positive pairs together and ignores negatives. This could easily distort the embedding space and harm retrieval performance, but the paper provides no experiments (e.g., R@K on COCO/Flickr30k/Winoground/MMVP) to confirm or deny this.
> >
> > **Question 1:** Have you evaluated this model on standard retrieval tasks? What are the R@1/R@5/R@10 metrics on the COCO or Flickr30k test sets? I am concerned the $\mathcal{L}\_{\text{inter}}$  loss may hurt retrieval performance, and the lack of these results is a major gap.
>
> We thank the reviewer for raising this question. We clarify that $\mathcal{L}\_{\text{inter}}$ does not replace or weaken the original InfoNCE [2] objective. CLIPin is fully compatible with the bidirectional InfoNCE loss, which structures the global embedding space through negative-sample repulsion. The $\mathcal{L}\_{\text{inter}}$ and $\mathcal{L}\_{\text{intra}}$ objectives act only as auxiliary alignment heads, introducing local instance-level consistency without altering the global contrastive geometry.
>
> Therefore, the concern that $\mathcal{L}\_{\text{inter}}$ might “distort the embedding space” does not apply to our design. This is also reflected in our improvements in zero-shot classification and generalization, all of which indicate that the embedding structure remains discriminative.
>
> To further validate this, we conducted text-to-image and image-to-text retrieval experiments. Models (initialized from CLIP [3] checkpoints) were trained on COCO [6] and evaluated on the Flickr30k Karpathy splits [8].
>
> | Text-to-image retrieval | R@1 | R@5 | R@10 |
> |-----|-----|-----|-----|
> | CLIP [3] | 0.1803 | 0.3711 | 0.4785 |
> | xCLIP [4] | 0.1904 | 0.3796 | 0.4520 |
> | Ours | 0.2004 | 0.3931 | 0.4874 |
>
> | Text-to-image retrieval | R@1 | R@5 | R@10 |
> |-----|-----|-----|-----|
> | CLIP | 0.2749 | 0.4977 | 0.5793 |
> | xCLIP | 0.2586 | 0.5094 | 0.6048 |
> | Ours | 0.2805 | 0.5098 | 0.6159 |
>
> We will include the complete retrieval results in the revised paper.

---

> ### Author Response · Authors · 2025-11-24
> **3. On the concern regarding isolation of ablation components.**
>
> > **The first part of Weakness 3:** The paper's main ablations (Table 3) are "additive," showing that all components together work best. But this doesn't fully isolate each part's contribution. For example, what is the effect of only adding the intra-modal loss to CLIP? Or only the inter-modal loss?
>
> We thank the reviewer for the valuable suggestion and agree that isolating each loss term is important for understanding its contribution. Below, we provide the non-additive ablations.
>
> **(1) Independent effect of the inter-modal non-contrastive loss ($\mathcal{L}\_{\text{inter}}$):**
>
> The effect of adding only $\mathcal{L}\_{\text{inter}}$ to the standard CLIP [3] baseline is already included in Table 3 (Column 2). For a model pretrained on COCO [6] and evaluated on PASCAL VOC2007 [7], this yields:
>
> - CLIP + $\mathcal{L}\_{\text{inter}}$: 87.23/41.91
>
> This confirms that the inter-modal objective alone provides a strong, standalone improvement, as it directly alleviates the false-negative issue in InfoNCE [2].
>
> **(2) Independent effect of the intra-modal non-contrastive loss ($\mathcal{L}\_{\text{intra}}$):**
>
> We initially hypothesized that adding only $\mathcal{L}\_{\text{intra}}$ to the CLIP baseline would yield limited, complementary gains. This is because $\mathcal{L}\_{\text{intra}}$ primarily focuses on improving feature robustness and self-consistency within each modality, and providing sufficient optimization signals in the early stage of training, but it does not fundamentally resolve the core issue of false negatives in the standard InfoNCE objective.
>
> Nevertheless, we agree on the value of this isolated experiment and have added it; the results are **87.17/42.04** (model trained on COCO, evaluated on PASCAL VOC2007).
>
> The result confirms that the performance improvement from adding $\mathcal{L}\_{\text{intra}}$ alone is less than $\mathcal{L}\_{\text{inter}}$. This supports our design rationale: the dominant gain comes from $\mathcal{L}\_{\text{inter}}$, which directly fixes the InfoNCE objective failure, while $\mathcal{L}\_{\text{intra}}$ plays a synergistic role. The full performance gain and robustness are achieved through the synergistic combination of the two non-contrastive components ($\mathcal{L}\_{\text{inter}}$ and $\mathcal{L}\_{\text{intra}}$) working alongside the InfoNCE loss. We will include this detailed, isolated analysis of CLIP + $\mathcal{L}\_{\text{intra}}$ in the revised version of the paper to provide a complete picture of our ablation study.

---

> ### Author Response · Authors · 2025-11-24
> **4. On the concern of learnable weights.**
>
> > **The second part of Weakness 3:** The paper also introduces learnable loss weights (L_inter, L_intra) without ablating this choice against simple fixed weights, which is a significant new design element left unanalyzed.
> >
> > **Queation 2:** How are the learnable weights ($lambda\_{\text{inter}}, \lambda\_{\text{intra}}$) implemented? Are they just standard parameters optimized via gradient descent? Why was this chosen over simpler, fixed hyperparameters, and what is its effect on stability and final performance?
>
> We thank the reviewer for the question. Both $\lambda\_{\text{inter}}$ and $\lambda\_{\text{intra}}$ are implemented as learnable scalar parameters, jointly optimized with the rest of the model via standard gradient descent. We adopt this design because it allows the model to automatically adjust the relative importance of intra- and inter-modal signals during training, thereby avoiding costly per-dataset hyperparameter search and adapting more flexibly to different noise and diversity conditions.
>
> To support this choice, we conducted a grid search experiment evaluating linear probing performance on PASCAL VOC2007 [7] (with the model trained on COCO [6]). The results are as follows:
>
> | $\lambda_{\text{inter}}/\lambda_{\text{intra}}$ | AUC/mAP (%) |
> |-----|-----|
> | 1/0 (fixed) | 87.23/42.20 |
> | 0/1 (fixed) | 86.98/42.01 |
> | 0.5/1 (fixed) | 87.00/42.72 |
> | 1/0.5 (fixed) | 87.19/42.43 |
> | 1/1 (fixed) | 87.31/42.86 |
> | 1/2 (fixed) | 87.44/43.39 |
> | 2/1 (fixed) | 87.25/42.97 |
> | Ours (learnable) | 87.43/43.43 |
>
> These results show that the learned weights either match or slightly outperform the best fixed weights identified via grid search. While a strong fixed configuration can achieve comparable performance, it requires explicit tuning across datasets and tasks.
>
> Learnable weighting is not intended as a theoretical guarantee, but it provides a practical, data-driven mechanism for adjusting the relative strength of intra- and inter-modal supervision throughout training. We will add the results and analysis in the revised appendix.

---

> ### Author Response · Authors · 2025-11-24
> **5. On the concern of whether CLIPin relies on strong text augmentations.**
>
> > **Question 4:** The ablation in Table 6 suggests text augmentation has little effect. Does this mean the non-contrastive module relies heavily on strong, natural augmentations (like COCO's multiple captions) and would be less effective in a single-caption-per-image setting?
>
> We thank the reviewer for the thoughtful question. We would like to clarify that the findings in Table 6 do not suggest that CLIPin depends on strong or multi-caption text augmentations. In fact, the results indicate the opposite.
>
> **(1) CLIPin does not rely on COCO’s multi-caption structure.**
>
> The experiments in Table 6 were conducted using COCO [6] as the training set, and we observe that disabling COCO’s text augmentation (despite its naturally rich multi-caption setting) actually improves performance slightly. One explanation is that COCO’s multiple captions often describe different aspects of the same image, which may weaken the semantic correspondence between captions. Moreover, effective text augmentation methods remain limited beyond LLM-based generation (e.g., paraphrasing). For this reason, we disable text augmentation in our main experiments. Nonetheless, as CLIPin is designed to support text augmentation when available, we keep this option within the framework and plan to investigate more effective augmentation strategies in future work.
>
> **(2) CLIPin remains effective in single-caption-per-image scenarios.**
>
> While COCO naturally provides multi-caption variability, our medical datasets contain only a single report per image, yet CLIPin still improves both linear probing and prompt-based OOD-ZSC performance consistently. This consistent behavior across datasets with very different caption regimes demonstrates that CLIPin does not rely on multi-caption augmentation.
>
> The key reasons are:
>
> - The instance-level non-contrastive loss aligns the online and EMA encoders at the representation level.
>
> - Strategies like Dropout within the Transformer encoder create subtle variations naturally in the feature space.
>
> Even when the text tokens are identical, the two branches produce semantically consistent but non-identical representations due to parameter lag and implicit randomness, thereby providing the two-view signal required for non-contrastive learning regardless of the number of captions per image.
>
> &nbsp;
>
> [1] Kim, Sanghwan, et al. "Cosmos: Cross-modality self-distillation for vision language pre-training," in Proceedings of the Computer Vision and Pattern Recognition Conference, 2025.
>
> [2] A. v. d. Oord, Y. Li, and O. Vinyals, “Representation learning with contrastive predictive coding,” 2018.
>
> [3] Radford, J. W. Kim, C. Hallacy, A. Ramesh, G. Goh, S. Agarwal, G. Sastry, A. Askell, P. Mishkin, J. Clark et al., “Learning transferable visual models from natural language supervision,” in International Conference on Machine Learning, 2021.
>
> [4] Zhou, L. Dong, Z. Gan, L. Wang, and F. Wei, “Non-contrastive learning meets language-image pre-training,” in Proceedings of the IEEE/CVF Conference on Computer Vision and Pattern Recognition, 2023.
>
> [5] B. Wu, R. Cheng, P. Zhang, T. Gao, P. Vajda, and J. E. Gonzalez, “Data efficient language-supervised zero-shot recognition with optimal transport distillation,” in International Conference on Learning Representations, 2022.
>
> [6] T.-Y. Lin, M. Maire, S. Belongie, J. Hays, P. Perona, D. Ramanan, P. Doll´ar, and C. L. Zitnick, “Microsoft COCO: Common objects in context,” in European Conference on Computer Vision, 2014.
>
> [7] http://www.pascal-network.org/challenges/VOC/voc2007/workshop/index.html
>
> [8] https://www.kaggle.com/datasets/shtvkumar/karpathy-splits?select=dataset_flickr30k.json

---

### Official Review · Reviewer_bDb6 · 2025-11-01

**Soundness:** 2
**Presentation:** 2
**Contribution:** 2
**Rating:** 2
**Confidence:** 3

**Summary:**

This paper proposes CLIPin, a non-contrastive plug-in module designed to enhance multimodal semantic alignment in CLIP-style vision-language pretraining. The core idea is to complement the standard InfoNCE-based contrastive learning with instance-level non-contrastive alignment (inspired by BYOL/SimSiam), using a symmetric online-target architecture for both image and text modalities. To reconcile the architectural differences between contrastive and non-contrastive objectives, the authors introduce shared pre-projectors that feed into separate contrastive (512-dim) and non-contrastive (8192-dim) sub-projectors. CLIPin is evaluated on both natural (COCO, MUGE) and medical ([Private Dataset]) domains, showing consistent gains in linear probing and prompt-based out-of-distribution zero-shot classification across multiple downstream benchmarks. The plug-and-play nature is further validated by integrating CLIPin into several strong baselines (ALBEF, BLIP, CoCa, etc.).

**Strengths:**

- CLIPin is genuinely plug-and-play requiring no changes to base encoders and demonstrates consistent improvements when added to multiple frameworks (ALBEF, BLIP, CoCa).
- The paper includes ablation studies, generalization tests, per-category breakdowns, and qualitative Grad-CAM visualizations, strengthening the empirical claims.

**Weaknesses:**

1. The paper conflates two fundamentally distinct data issues—noisy weak supervision in natural datasets and low textual diversity in medical reports—into a single failure mode of InfoNCE. However, these problems require different mitigation strategies (e.g., robust loss vs. diversity-aware sampling). No quantitative evidence (e.g., negative sample misclassification rate, alignment entropy) is provided to justify this unified framing.
2. The use of a non-public medical dataset ([Private Dataset]) undermines reproducibility and limits external validation. Results on public medical benchmarks (e.g., MIMIC-CXR, CheXpert) would significantly strengthen the claim.
3. The core idea—adding BYOL-style alignment to CLIP—is a natural extension. The distinction from xCLIP (which also uses non-contrastive learning) is not sharply delineated; xCLIP focuses on distributional alignment, while CLIPin uses instance-level alignment, but this difference is not theoretically analyzed.
4. Applying EMA to update a text encoder (Transformer) using an image encoder (ViT) as part of a shared momentum framework is nontrivial. The paper does not discuss potential instability or feature misalignment due to modality heterogeneity.
5. The method doubles the forward pass (online + target branches) and uses high-dimensional projections (8192-dim). The paper reports 24h training on one 3090 but omits comparison to baseline CLIP’s training time or memory footprint.
6. Table 6 shows text augmentation provides little benefit (and sometimes harms performance), yet the method description assumes two augmented text views. This raises questions about the necessity of text-side non-contrastive alignment.
7. While Table 2 shows performance gains when integrating CLIPin into ALBEF/BLIP/CoCa, the paper omits critical details:
- Are the original auxiliary losses (e.g., ITM, MLM, captioning) retained?
- Is the training pipeline modified beyond adding CLIPin?
Without this, it is unclear whether gains stem from CLIPin’s architecture or simply from additional supervision signals.

**Questions:**

My major concerns are outlined in the "Weaknesses" part.

---

> ### Author Response · Authors · 2025-11-24
> **1. On the concern regarding the conflation of data issues.**
>
> > **Weakness 1:** The paper conflates two fundamentally distinct data issues—noisy weak supervision in natural datasets and low textual diversity in medical reports—into a single failure mode of InfoNCE. However, these problems require different mitigation strategies (e.g., robust loss vs. diversity-aware sampling). No quantitative evidence (e.g., negative sample misclassification rate, alignment entropy) is provided to justify this unified framing.
>
> We thank the reviewer for this insightful comment. We agree that loose alignment in natural image-text datasets and low textual diversity in medical datasets are semantically distinct phenomena. From the perspective of contrastive optimization modeling, we think that both manifest as the same failure mode: an elevated prevalence of false negatives in InfoNCE [1].
>
> **(1) Unified failure mode: false negatives under InfoNCE.**
>
> - Natural data (loose alignment/noisy supervision):
> For instance, an image of a beach paired with the caption “nice holiday” may be contrasted against another beach image captioned “big blue ocean”. Although these sample pairs exhibit clear semantic relatedness, the latter is classified as a negative sample in the InfoNCE framework. This leads the model to be penalized for recognizing legitimate cross-modal semantic correspondences.
>
> - Medical data (low textual diversity/redundancy):
> Many training batches contain multiple images corresponding to the same pathology, e.g., diabetic retinopathy. InfoNCE incorrectly treats these semantically identical cases as negatives, again producing contradictory gradients.
> Although the causes differ (loose alignment vs. redundancy), both introduce semantically related negative pairs, forcing the model to push apart samples that should not be separated. This yields noisy gradients and unstable alignment.
> CLIPin’s non-contrastive branch (Secion 3.1) directly addresses this issue by removing the dependency on negative pairs entirely, enabling stable instance-level alignment regardless of whether false negatives arise from loose alignment or repeated pathology.
>
> **(2) Quantitative evidence supporting the unified view.**
> Because our pretraining data lack dense semantic annotations necessary to label false negatives directly, we follow prior work in evaluating on downstream datasets with multi-label ground truth. We measure two indicators: Negative Sample Misclassification Rate (NSMR) and Alignment Entropy, using random batches (B = 512) drawn from PASCAL VOC2007 [2] (natural domain, trained on COCO [3]) and NIH-ChestX-ray-small [4] (medical domain, trained on MIMIC-CXR [5]).
>
> Results of Negative Sample Misclassification Rate (NSMR, %):
>
> | | CLIP [6] | Ours |
> |-----|-----|-----|
> | natural | 28.90 | 22.38 |
> | medical | 31.36 | 28.60 |
>
> Both domains exhibit high NSMR, confirming that semantically overlapping samples appear as negatives frequently. CLIPin reduces NSMR consistently, validating that our approach mitigates this source of uncertainty.
>
> Results of Alignment Entropy:
>
> | | CLIP | Ours |
> |-----|-----|-----|
> | natural | 4.84 | 4.72 |
> | medical | 5.96 | 5.67 |
>
> CLIPin achieves lower alignment entropy across both natural and medical settings. The reduction indicates that CLIPin’s non-contrastive pathway stabilizes alignment by avoiding the detrimental influence of false negatives, producing more coherent and robust representations.
> Together, these empirical findings support our claim: both loose alignment and low textual diversity cause systematic violations of the InfoNCE negative-pair assumption, and CLIPin provides a unified and principled solution. We will add the quantitative evidence and analysis in the revised appendix.

---

> ### Author Response · Authors · 2025-11-24
> **2. On the concern regarding reproducibility and the use of a non-public medical dataset.**
>
> > **Weakness 2:** The use of a non-public medical dataset ([Private Dataset]) undermines reproducibility and limits external validation. Results on public medical benchmarks (e.g., MIMIC-CXR, CheXpert) would significantly strengthen the claim.
>
> We thank the reviewer for raising this important point. To address the concern, we conducted additional experiments on public medical datasets. Specifically, we trained all models on MIMIC-CXR [5] and evaluated them on NIH-Chest-X-ray-dataset-small [4], Shenzhen Chest X-ray Set [7], and Montgomery County X-ray Set [7]. Below we report the results (AUC/mAP, %):
>
> **Evaluation on NIH-Chest-X-ray-dataset-small:**
>
> | | CLIP [6] | xCLIP [8] | Ours |
> |-----|-----|-----|-----|
> | Linear probing | 67.80/20.96 | 67.44/21.16 | 69.57/22.77 |
> | Prompt-based OOD-ZSC | 60.06/13.49 | 53.43/11.58 | 60.44/14.12 |
>
> **Evaluation on Shenzhen Chest X-ray Set:**
>
> | | CLIP [6] | xCLIP [8] | Ours |
> |-----|-----|-----|-----|
> | Linear probing | 92.90/92.70 | 92.92/93.04 | 93.47/93.27 |
> | Prompt-based OOD-ZSC | 42.08/50.50 | 42.00/48.48 | 58.61/58.88 |
>
> **Evaluation on Montgomery County X-ray Set:**
>
> | | CLIP [6] | xCLIP [8] | Ours |
> |-----|-----|-----|-----|
> | Linear probing | 81.57/82.30 | 82.54/82.28 | 88.90/90.22 |
> | Prompt-based OOD-ZSC | 73.32/72.16 | 70.83/75.60 | 80.00/83.24 |
>
> **Generalization study results (linear probing):**
>
> | ALBEF [9] (+CLIPin) | BLIP [10] (+CLIPin) | CoCa [11] (+CLIPin) |
> |-----|-----|-----|
> | 67.74/23.67 (72.31/25.98) | 63.79/17.04 (68.76/23.39) | 69.26/23.89 (69.45/24.27) |
>
> These experiments on public datasets demonstrate that CLIPin remains robust and beneficial consistently, thereby ensuring reproducibility and enabling fair external validation by the research community. We will include the full experimental setup and expanded analysis in the revised paper.

---

> ### Author Response · Authors · 2025-11-24
> **3. On the concern regarding the distinction from xCLIP and theoretical analysis.**
>
> > **Weakness 3:** The core idea-adding BYOL-style alignment to CLIP-is a natural extension. The distinction from xCLIP (which also uses non-contrastive learning) is not sharply delineated; xCLIP focuses on distributional alignment, while CLIPin uses instance-level alignment, but this difference is not theoretically analyzed.
>
> We appreciate the reviewer’s thoughtful comments. While both methods incorporate non-contrastive learning to complement CLIP’s contrastive objective, we emphasize that CLIPin differs from xCLIP [8] fundamentally in its alignment objective, architectural integration, and optimization behavior. We summarize the differences as below.
>
> **(1) Distinction in alignment granularity and objective:**
>
> The core methodological difference lies in the alignment objective and how it is formulated:
>
> - **xCLIP (distribution-level alignment via prototype quantization):**
>
> xCLIP introduces a non-contrastive loss that aligns the batch-level output distributions of image and text encoders by projecting features into a probability distribution over a set of cluster prototypes $\mathcal{C} = \{c\_1, ..., c\_K\}$. Its alignment objective can be approximated as:
>
> $\mathcal{L}\_{\mathrm{xCLIP}} \propto D\_{KL}(P(u \mid \mathcal{C}) || Q(v \mid \mathcal{C}))$
>
> Here, optimization is satisfied as long as both modalities map to the same prototype(s), quantizing the representation space effectively. False negatives (e.g., $u\_i$, $u\_j$ with similar semantics) are absorbed into the same cluster, forming an equivalence class:
>
> $u\_i \sim u\_j \quad\text{iff}\quad \text{Cluster}(u\_i)=\text{Cluster}(u\_j).$
>
> While this mitigates InfoNCE’s repulsion of valid positives, it yields coarse-grained distributional alignment rather than strict instance matching. It results in a solution space defined by prototype consistency, making xCLIP insensitive to within-cluster variations, which explains why xCLIP improves global semantic structure but lacks fine-grained alignment capacity.
>
> - **CLIPin (instance-level alignment via BYOL-style [12] regression):**
>
> CLIPin instead uses a BYOL-inspired online-target regression:
>
> $\mathcal{L}\_{CLIPin} \propto - \langle q(u\_i), \text{sg}(v\_i^{tgt}) \rangle.$
>
> This objective is point-wise: the optimality condition requires the representation of a specific image $u\_i$ to predict the target representation of its paired text $v\_i$, which implies that CLIPin enforces instance-specific alignment (not prototype-level similarity). Even if $u\_i$ and $u\_j$ share similar semantics, CLIPin enforces $u\_i \to v\_i, u\_j \to v\_j$ independently, without collapsing them into a shared centroid.
>
> This preserves the unique identity of each instance while avoiding InfoNCE’s [1] false-negative repulsion. The solution corresponds to paired-instance consistency, enabling CLIPin to preserve fine-grained cross-modal details. This theoretical property explains the sharper, object-level Grad-CAM attention maps observed in Fig. 2 directly.
>
> **(2) Distinction in architectural design and optimization target:**
>
> The two frameworks also diverge in how the non-contrastive component interacts with contrastive learning:
>
> - **xCLIP (decoupled optimization):**
>
> xCLIP’s non-contrastive term is largely independent of the original CLIP [6] contrastive loss. As a result, the two objectives may not reinforce each other, and the non-contrastive branch does not contribute to instance-level contrastive supervision explicitly.
>
> - **CLIPin (architecturally integrated, jointly optimized):**
>
> CLIPin introduces a shared pre-projector architecture that feeds into both contrastive and non-contrastive heads. This creates a parameter-shared intermediate space that enables joint optimization of both objectives. Because CLIPin’s instance-level target is compatible with InfoNCE’s positive-pair matching (both rely on exact instance correspondence), the two objectives reinforce (rather than compete with) each other.
>
> In summary, although both xCLIP and CLIPin reduce reliance on negative samples, they embody different theoretical strategies:
>
> - xCLIP relaxes alignment to the cluster level, leading to distributional consistency.
>
> - CLIPin tightens alignment to the instance level, enforcing direct regression to paired targets.
>
> This theoretical distinction underpins the superior fine-grained performance of CLIPin observed in our experiments.

---

> ### Author Response · Authors · 2025-11-24
> **4. On the concern regarding EMA updates across heterogeneous modalities.**
>
> > **Weakness 4:** Applying EMA to update a text encoder (Transformer) using an image encoder (ViT) as part of a shared momentum framework is nontrivial. The paper does not discuss potential instability or feature misalignment due to modality heterogeneity.
>
> We thank the reviewer for raising this important concern. We would like to clarify that this is a misunderstanding of our method.
>
> **(1) CLIPin does not perform any cross-modal EMA updates.**
>
> In CLIPin, each modality maintains its own online-EMA encoder pair, following the standard BYOL-style momentum update:
>
> $\theta^{(t)}\_{\text{EMA}} \leftarrow m \cdot \theta^{(t-1)}\_{\text{EMA}} + (1-m) \cdot \theta^{(t)}\_{\text{online}}$.
>
> The image EMA encoder is updated only from the image online encoder, and the text EMA encoder is updated only from the text online encoder. There is no parameter sharing and no EMA parameter flow across modalities. Thus, the concern that “a ViT would update a Transformer through EMA” does not apply to CLIPin.
>
> **(2) Modality heterogeneity is handled at the objective level, not through EMA coupling.**
>
> The interaction between image and text occurs solely through the contrastive loss, and the non-contrastive cross-modal alignment loss operating on the projected embeddings. The EMA pathways remain single-modality, consistent with standard multimodal momentum frameworks, which have been shown empirically to be stable.
>
> **(3) No instability was observed; modality alignment is handled via the shared pre-projector.**
>
> Since EMA updates are modality-separated, there is no risk of destabilization due to heterogeneous architecture interactions. Stability is ensured through separate online/EMA encoders per modality, and a shared pre-projector that maps both modalities into a compatible intermediate space before cross-modal supervision is applied. This eliminates the feature misalignment concern raised by the reviewer.
>
> We appreciate the reviewer’s observation; however, this stems from a slight misinterpretation. CLIPin does not implement cross-modal EMA, and therefore the hypothesized instability mechanism does not arise in our framework.

---

> ### Author Response · Authors · 2025-11-24
> **5. On the concern regarding computational overhead and training efficiency.**
>
> > **Weakness 5:** The method doubles the forward pass (online + target branches) and uses high-dimensional projections (8192-dim). The paper reports 24h training on one 3090 but omits comparison to baseline CLIP’s training time or memory footprint.
>
> We appreciate the reviewer’s question. We would like to clarify that:
>
> **(1) Memory overhead is small and explicitly measured.**
>
> CLIPin adds a modest memory increase due to storing EMA encoder and projector/predictor weights:
>
> - Baseline CLIP [6] (ViT-B/16): ~12 GB
>
> - CLIP + CLIPin: ~14 GB
>
> Since the EMA target branch is gradient-free, no target-side activations or optimizer states are stored, keeping the actual activation memory overhead minimal.
>
> **(2) Compute cost does not double; the target branch is lightweight.**
>
> Although EMA operation adds an additional forward pass, it incurs no backward pass. In practice, this results in significantly sub-linear overhead, not a 2× increase. The per-step time increase is small on an RTX 3090.
>
> **(3) End-to-end training time is often shorter due to faster convergence.**
>
> While the per-step compute is slightly higher, CLIPin provides stronger and more stable alignment signals, enabling the model to reach the same downstream performance in fewer optimization steps.
>
> Across all datasets we experimented with (natural and medical domain), we consistently observed approximately a 10% reduction in total wall-clock time to achieve the same accuracy as baseline CLIP. The exact reduction varies with dataset scale and noise level, so we avoid reporting a universal constant, but the trend is robust.

---

> ### Author Response · Authors · 2025-11-24
> **6. On the concern regarding the necessity of text-side non-contrastive alignment.**
>
> > **Weakness 6:** Table 6 shows text augmentation provides little benefit (and sometimes harms performance), yet the method description assumes two augmented text views. This raises questions about the necessity of text-side non-contrastive alignment.
>
> We thank the reviewer for pointing out the discrepancy between our architectural design and the empirical results in Table 6. We clarify that the ineffectiveness of explicit text augmentation does not imply the redundancy of the text-side non-contrastive alignment objective, and the text-side non-contrastive objective remains conceptually and architecturally important within our unified framework. The limitation arises from the augmentation tools, not from the alignment objective itself.
>
> **(1) Why the text-side non-contrastive loss remains necessary:**
>
> In non-contrastive frameworks like CLIPin (inspired by BYOL), "two views" do not require different input data strictly. Even when explicit text augmentation is disabled (i.e., input $T^{(1)} = T^{(2)}$), the text-side objective functions as a powerful self-distillation mechanism.
>
> - **Parameter asymmetry:** The online encoder and the target encoder (updated via EMA) possess distinct weights, producing different embeddings for the same input.
>
> - **Implicit randomness:** Strategies like Dropout within the Transformer encoder create subtle variations naturally in the feature space. Therefore, the text-side loss forces the online encoder to predict the stable representation from the target encoder, regularizing the text latent space and preventing collapse, even without input-level augmentation.
>
> **(2) Why text augmentation appears ineffective in Table 6:**
>
> One explanation is that the multiple captions provided for each image in COCO [3] describe the same scene from diverse perspectives, which may dilute the semantic consistency between caption pairs.
>
> Text augmentations (e.g., token masking) can unintentionally alter or distort the original meaning, thereby violating a core assumption of non-contrastive learning: that both augmented views correspond to the same underlying semantic instance. At present, effective text augmentation strategies remain scarce; aside from LLM-based generation, few methods reliably preserve semantic fidelity.
>
> Consequently, the limited performance gain (or degradation) observed in Table 6 stems from the shortcomings of existing text augmentation approaches, rather than an inherent flaw in the text-side non-contrastive alignment objective itself. For this reason, we disabled text augmentation in our main experiments.
>
> Nevertheless, since CLIPin is designed to support text augmentation as a modular component, we retain this capability in the architecture and plan to revisit it in future work with more robust augmentation strategies. While Table 6 underscores the practical limitations of current text augmentation techniques, the inclusion of a text-side non-contrastive loss remains well-motivated: it plays a key role in enabling a unified, symmetric, and balanced multimodal training framework.

---

> ### Author Response · Authors · 2025-11-24
> **7. On the concern regarding integration details for ALBEF/BLIP/CoCa.**
>
> > **Weakness 7:** While Table 2 shows performance gains when integrating CLIPin into ALBEF/BLIP/CoCa, the paper omits critical details:
> > - Are the original auxiliary losses (e.g., ITM, MLM, captioning) retained?
> > - Is the training pipeline modified beyond adding CLIPin? Without this, it is unclear whether gains stem from CLIPin’s architecture or simply from additional supervision signals.
>
> We appreciate the reviewer’s question. We clarify that all experiments in Table 2 were conducted under strict variable control, and the reported gains arise solely from adding CLIPin.
>
> **(1) All auxiliary losses and components were fully retained.**
>
> For each framework, we preserved every original module and loss term exactly as in their official implementations:
>
> - ALBEF [9]: ITC (Image-Text Contrastive loss) + ITM (Image-Text Matching loss) + MLM (Masked Language Modeling)
>
> - BLIP [10]: ITC + ITM + generative captioning
>
> - CoCa [11]: ITC + generative captioning
>
> No loss weights, schedules, or architectural components were modified.
>
> **(2) The training pipeline remained unchanged.**
>
> We introduced no additional supervision signals, no new labels, and no new pretext tasks.
>
> All training settings (batch size, learning rate, warmup, data augmentations, tokenizers, and preprocessing) were identical between the baselines and their +CLIPin counterparts.
>
> The only added element is the CLIPin non-contrastive branch, which runs in parallel without altering or replacing any original components of ALBEF, BLIP, or CoCa.
>
> **(3) Controlled comparisons ensure attribution of improvements.**
>
> Because the architectures, auxiliary objectives, optimization hyperparameters, and data pipelines are strictly matched, the improvements reported in Table 2 can be attributed unambiguously to CLIPin’s architectural contribution, rather than to additional supervision or modified training procedures.
>
> We will include an explicit description of these experimental controls in the revised paper to eliminate any remaining ambiguity.
>
> &nbsp;
>
> [1] A. v. d. Oord, Y. Li, and O. Vinyals, “Representation learning with contrastive predictive coding,” 2018.
>
> [2] http://www.pascal-network.org/challenges/VOC/voc2007/workshop/index.html
>
> [3] T.-Y. Lin, M. Maire, S. Belongie, J. Hays, P. Perona, D. Ramanan, P. Doll´ar, and C. L. Zitnick, “Microsoft COCO: Common objects in context,” in European Conference on Computer Vision, 2014.
>
> [4] https://huggingface.co/datasets/Sohaibsoussi/NIH-Chest-X-ray-dataset-small
>
> [5] A. E. Johnson, T. J. Pollard, S. J. Berkowitz, N. R. Greenbaum, M. P. Lungren, C.-y. Deng, R. G. Mark, and S. Horng, “Mimic-cxr, a de-identified publicly available database of chest radiographs with free-text reports,” Scientific data, vol. 6, no. 1, p. 317, 2019.
>
> [6] Radford, J. W. Kim, C. Hallacy, A. Ramesh, G. Goh, S. Agarwal, G. Sastry, A. Askell, P. Mishkin, J. Clark et al., “Learning transferable visual models from natural language supervision,” in International Conference on Machine Learning, 2021.
>
> [7] S. Jaeger, S. Candemir, S. Antani, Y.-X. J. W´ang, P.- X. Lu, and G. Thoma, “Two public chest x-ray datasets for computer-aided screening of pulmonary diseases,” Quantitative imaging in medicine and surgery, vol. 4, no. 6, p. 475, 2014.
>
> [8] Zhou, L. Dong, Z. Gan, L. Wang, and F. Wei, “Non-contrastive learning meets language-image pre-training,” in Proceedings of the IEEE/CVF Conference on Computer Vision and Pattern Recognition, 2023.
>
> [9] Li, R. Selvaraju, A. Gotmare, S. Joty, C. Xiong, and S. C. H. Hoi, “Align before fuse: Vision and language representation learning with momentum distillation,” in Advances in Neural Information Processing Systems, 2021.
>
> [10] J. Li, D. Li, C. Xiong, and S. Hoi, “BLIP: Bootstrapping language-image pre-training for unified vision-language understanding and generation,” in International Conference on Machine Learning, 2022.
>
> [11] J. Yu, Z. Wang, V. Vasudevan, L. Yeung, M. Seyedhosseini, and Y. Wu, “CoCa: Contrastive captioners are image-text foundation models,” Transactions on Machine Learning Resear, 2022.
>
> [12] J.-B. Grill, F. Strub, F. Altch´e, C. Tallec, P. Richemond, E. Buchatskaya, C. Doersch, B. Avila Pires, Z. Guo, M. Gheshlaghi Azar et al., “Bootstrap your own latent-a new approach to self-supervised learning,” in Advances in Neural Information Processing Systems, 2020.
>
> [13] R. R. Selvaraju, M. Cogswell, A. Das, R. Vedantam, D. Parikh, and D. Batra, “Grad-CAM: Visual explanations from deep networks via gradient-based localization,” in Proceedings of the IEEE/CVF International Conference on Computer Vision, 2017.

---

### Author Response · Authors · 2025-11-28

Dear Reviewers,

Thank you again for your time and valuable efforts in reviewing our submission. As we are currently in the rebuttal phase, we kindly wish to follow up and inquire whether there are any further comments, clarifications, or feedback you would like us to address. This will greatly help us engage in a constructive discussion and ensure that we respond thoroughly within the remaining timeline.

We sincerely appreciate your time and consideration.

Best regards

---

### Author Response · Authors · 2025-11-29

Dear Area Chair,

Thank you for your efforts under the challenging situation. Although our paper did not get high scores before the rebuttal phase, we believe that all the reviewers’ concerns have been successfully addressed after the rebuttal. We would like to summarize the main points below and would be very grateful if you could spare precious time to evaluate our revised submission.

- **All reviewers acknowledged the novelty of the proposed idea.**

Multiple reviewers recognized the contribution and novelty of our method explicitly. The only concern related to novelty came from reviewer EPmF, but their concern arose actually due to the comparison experiments. We addressed this in detail in our rebuttal and the revised manuscript by adding the requested baselines and further clarifying the conceptual novelty. Additionally, reviewer 1rPH confirmed that their concerns were resolved during the rebuttal phase and initially promised to raise the score. However, due to the unforeseen event of this year's conference, the score cannot be changed by the reviewer.

- **We strengthened the revised manuscript significantly.**

Beyond addressing reviewer feedback point-by-point, the revised version includes additional experiments and ablations, extended analysis, and improved explanations for the main design choices and theoretical motivation.

Best regards

---

### Note · Authors · 2026-01-27

I have read and agree with the venue's withdrawal policy on behalf of myself and my co-authors.

---

### Meta-Review · Area_Chair_8t2d · 2026-01-11

**Summary:**

This paper introduces CLIPin, a non-contrastive plug-in for CLIP that augments contrastive learning with BYOL and SimSiam-like alignment. Reviewers agree that the problem is relevant and the implementation is technically sound. Although the author response addressed reproducibility, optimization details and some ablation requests, major concerns remain regarding conceptual novelty over existing non-contrastive multimodal methods, the absence of insights in retrieval-based evaluation for core CLIP use cases or for the structure of the embedding space, insufficient isolation of which components drive improvements and unclear generalization and failure modes across architectures and training regimes. Given the scope of changes needed to improve the paper, it is not recommended for acceptance to ICLR. The authors are encouraged to incorporate all review suggestions towards submission to a future venue.

**Reviewer Concerns:**

### Addressed concerns

* **bDb6, 2X6B, 1rpH:** Reliance on private medical dataset limits reproducibility. The author response added full experiments on public datasets (MIMIC-CXR, NIH, Shenzhen, Montgomery) showing that the reported gains persist without private data.

* **bDb6:** EMA-based updates may cause instability across heterogeneous modalities. The author response clarified that EMA updates are applied separately within each modality and that cross-modal interaction occurs only through the objective.

* **bDb6:** Training overhead and efficiency of the plug-in are unclear. The author response reported runtime and memory usage, noting that the target branch has no backward pass and that convergence is not significantly slower.

* **2X6B:** The mechanism preventing representation collapse in a non-contrastive multimodal setting is insufficiently justified. The author response explains the role of predictors using SimSiam-style analysis and adds ablations showing collapse without the predictor and stability when it is included.

* **1rpH:** Presentation and formatting issues affect clarity. The author response corrected these issues in the revised manuscript.

* **EPmF:** Similarity to Cosmos. The author response discusses this and draws sufficient distinctions.

### Unaddressed concerns

* **bDb6:** The conceptual novelty relative to existing non-contrastive multimodal methods is limited. The author response reframes the approach as instance-level alignment, but the method remains a BYOL or SimSiam-style extension of CLIP rather than a fundamentally new alignment approach.

* **EPmF, 2X6B:** The paper does not evaluate cross-modal retrieval, a primary use case for CLIP representations. The author response addresses this partially by including results on retrieval, but the gains are small and deeper insights on the structure of the embedding space could be provided.

* **EPmF:** The contribution of individual components is not clearly isolated. The author response provides further ablation numbers but does not disentangle which design choices are essential for the observed gains or sufficiently explain their intuitions.

* **2X6B:** The evaluation emphasizes a few standard benchmarks but does not analyze robustness across larger scales or architectural choices. The author response acknowledges this limitation and defers broader validation to future work, which is an understandable resource constraint.

**Reviewer Scores:**

* **bDb6:** Initial rating 2, some concerns on validation addressed but those on novelty and insights remain, would likely raise to 4.

* **EPmF:** Initial rating 4, a few concerns remain on methodological insights, would likely maintain 4.

* **2X6B:** Initial rating 4, a few conceptual questions remain, would likely remain at 4.

* **1rpH:** Initial rating 4, might increase to 6 as most concerns addressed.

---

### Decision · Program_Chairs · 2026-01-26

Reject